# THE GENERALIZATION RIDGE: INFORMATION FLOW IN NATURAL LANGUAGE GENERATION

## ABSTRACT

Transformer-based language models have achieved state-of-the-art performance in natural language generation (NLG), yet their internal mechanisms for synthesizing task-relevant information remain insufficiently understood. While prior studies suggest that intermediate layers often yield more generalizable representations than final layers, how this generalization ability emerges and propagates across layers during training remains unclear. To address this gap, we propose InfoRidge, an information-theoretic framework, to characterize how predictive information—the mutual information between hidden representations and target outputs—varies across depth. Estimating this quantity enables us to trace the flow of task-relevant information throughout the model during training. Our experiments across various models and datasets reveal a consistent non-monotonic trend: predictive information peaks in upper-middle layers—forming a **generalization ridge**—before declining in final layers, reflecting a transition between generalization and memorization. To further investigate this phenomenon, we introduce residual scaling coefficients—trainable scalar parameters applied to each residual block—which serve as functional probes for assessing the relative importance of individual transformer layers. These coefficients reveal that, under distribution shift, models downweight final layers and increasingly rely on ridge layers, highlighting their role in generalization. Together, these findings offer new insights into the internal mechanisms of transformers and underscore the critical role of intermediate layers in supporting generalization.

## 1 INTRODUCTION

Transformer-based language models generate text by predicting tokens autoregressively, and they have achieved remarkable performance across a wide range of natural-language uses (Vaswani et al., 2017; Dong et al., 2022). Nevertheless, we lack a rigorous understanding of how these models acquire and synthesize task-relevant information during training.

A growing body of research has shown that intermediate layers in deep neural networks often surpass final layers in terms of representational quality and generalization performance (Liu et al., 2019b; Voita et al., 2019; Ansuini et al., 2019; Ahrens et al., 2023; Uselis & Oh, 2025). In language models, intermediate layers often encode richer semantic and more robust features than final layers (Fan et al., 2024; Jin et al., 2024; Skean et al., 2025). However, questions still remain: ***In NLG, how does information evolve across layers during training, and how are different layers of the network functionally organized to support generalization versus memorization?***

To investigate these questions, we propose *InfoRidge*, an information-theoretic framework, to analyze information flow in language models. Building on matrix-based mutual information estimation (Giraldo et al., 2014), our approach quantifies how predictive signals transform across layers during training. We center our analysis on two complementary quantities: *predictive information*, defined as the mutual information $I(Z_\ell; Y)$ between the hidden representation $Z_\ell$ at layer $\ell$ and the next-token label $Y$, reflecting how much task-relevant information is preserved; *incremental information gain*, denoted as $I(\Delta Z_\ell; Y)$, where $\Delta Z_\ell$ is the residual changes between successive layer $\ell$ and $\ell - 1$, measuring the additional predictive information introduced by each transformer block.

Using *InfoRidge*, we uncover a non-monotonic trend: predictive information rises through the early and middle layers, peaks in the upper-middle layers, and then declines in the later layers. We name

this phenomenon the ***generalization ridge***, where the model encodes the more generalizable task-relevant information. This ridge marks a structural division of labor: intermediate layers concentrate generalizable features that transfer across distributions, while later layers increasingly specialize in task-specific memorization. Incremental information gain further shows that ridge layers introduce the largest increases in predictive information, marking them as key contributors to the emergence of the ridge. This analysis directly connects the information peak to generalizable behavior and clarifies how generalization and memorization are distributed across depth.

To further validate this interpretation, we introduce *residual scaling coefficients*—learnable scalar parameters $\beta_\ell$ applied to each residual block—while keeping all other model weights frozen, drawing inspiration from prior work on layer-wise adaptation (Liu et al., 2019a; Menghani et al., 2024). A higher $\beta_\ell$ value indicates greater reliance on the corresponding layer's output during prediction. These coefficients act as functional probes, revealing how the model redistributes layer-wise importance under different data distribution. Under in-distribution training, deeper layers retain higher residual weights, reflecting the model's reliance on memorized, task-specific features. When evaluated under distribution shift, models reduce reliance on late layers and increase reliance on the ridge, further supporting its role in generalization.

To understand the formation of the generalization ridge, we analyze both attention patterns and model capacity. Attention analysis shows ridge layers attend to tokens that capture broadly useful features, aligning with the information peak. Beyond attention patterns, we also investigate the conditions under which the ridge emerges. Our depth ablation results show that the ridge only emerges beyond a certain depth threshold. Below this threshold, predictive information increases monotonically, indicating that sufficient capacity is a prerequisite for generalization ridge to emerge.

**Contributions.** Our work provides a unified perspective on how predictive information is structured across depth in transformer-based language models for natural language generation tasks:

1. Our work tracks the evolution of predictive information throughout training, establishing a clear connection between predictive information flow and generalization. It reveals a non-monotonic peak in the middle layers, which we refer to as the ***generalization ridge***. This pattern reflects a meaningful transition in representational focus and aligns with stronger generalization behavior.

2. We introduce InfoRidge, an information-theoretic framework that applies matrix-based mutual information estimation to autoregressive language models to analyze information flow.

3. We introduce residual scaling coefficients as trainable indicators of how models shift representational focus across layers during training, providing a causal, adaptive measure of generalization.

## 2 RELATED WORK

Understanding how information is encoded and transformed across layers has been studied through probing classifiers (Alain & Bengio, 2016), attention flow (Vig & Belinkov, 2019), and information-theoretic approaches such as the information bottleneck (Shwartz-Ziv & Tishby, 2017), mutual information estimation (Goldfeld, 2019), and matrix-based entropy (Giraldo et al., 2014), offering different lenses to quantify representational capacity, abstraction, and invariance across layers.

A growing body of research has shown that intermediate layers in deep networks often outperform final layers in terms of representational quality and task performance (Ansuini et al., 2019; Yosinski et al., 2014; Uselis & Oh, 2025; Ahrens et al., 2023; Ando et al., 2023). In language models, mid-depth layers tend to capture richer semantic or robust features than output layers (Liu et al., 2019b; Voita et al., 2019; Jin et al., 2024; Fan et al., 2024). Transformer representations have long been observed to follow a structured progression from syntactic to semantic information, as shown by classical probing studies on linguistic knowledge and the reconstruction of the NLP pipeline (Liu et al., 2019b; Tenney et al., 2019). These findings challenge the assumption that deeper layers always yield better representations.

This pattern holds across settings such as transfer learning (Yosinski et al., 2014), continual learning (Ahrens et al., 2023), and out-of-distribution generalization (Uselis & Oh, 2025). Furthermore, recent work has evaluated representation quality using entropy, curvature, and invariance (Skean et al., 2025), while other studies have examined embedding drift and representational geome-

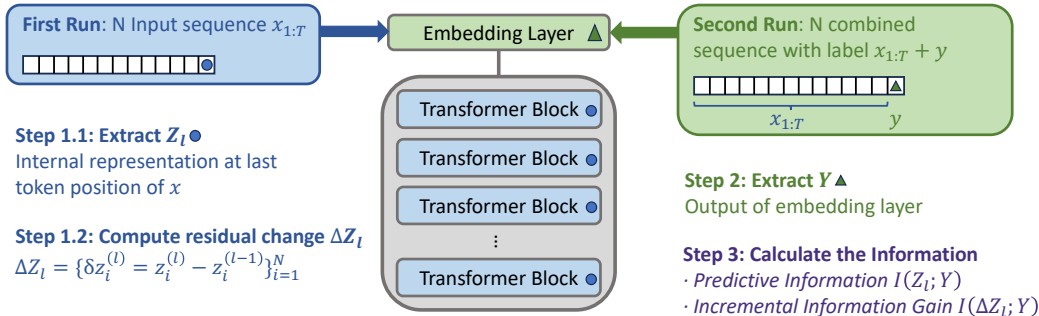

Figure 1: Overview of InfoRidge. (1) Step 1: Extract internal representations at each layer and compute residual changes between successive layers. (2) Step 2: Extract the target token embedding. (3) Step 3: Using these representations, we estimate the layer-wise predictive information $I(Z_\ell; Y)$ and the incremental information gain $I(\Delta Z_\ell; Y)$, which jointly characterize how information evolves across depth.

try (Merchant et al., 2020; Dar et al., 2022), analyzed memorization and factual recall (Haviv et al., 2022; Yu et al., 2023), and introduced causal perspectives on layer importance through mediation analysis and targeted interventions (Vig et al., 2020; Meng et al., 2022). Training-dynamics studies investigated how earlier models develop and refine semantic features across depth, providing an additional perspective on the evolution of layer-wise representations (Merchant et al., 2020; Kumar et al., 2023). Correlational probes (e.g., linear probes (Alain & Bengio, 2016)) measure only whether a feature can be decoded from a representation, which reflects correlation but not causal influence. In contrast, causal methods intervene on internal activations to test how changing a component alters the model's prediction, thereby identifying true causal contribution rather than mere feature presence. Our residual-scaling approach aligns with this causal perspective at the layer level.

However, the underlying causes and functional role of this phenomenon remain only partially understood, motivating further investigation. Our work addresses this gap by tracing the evolution of predictive information throughout training and establishing a clear connection between predictive information flow and generalization. We reveal a consistent non-monotonic peak in the middle layers—termed the *generalization ridge*—that reflects a meaningful transition in representational focus and aligns with stronger generalization behavior. Additionally, unlike prior work focused on classification tasks, we extend the analysis of information flow to generation tasks by quantifying the mutual information between hidden states and the next token. This enables us to understand how generalization and memorization dynamics evolve for next-token generation setting during training from an information-theoretic perspective.

## 3 INFORIDGE: INFORMATION ESTIMATION FRAMEWORK

We propose InfoRidge, an information-theoretic framework that uses mutual information to quantify how predictive information propagates through transformers layers during training in NLG.

**Motivating Insight.** Prior work has shown that internal representations in deep neural networks tend to align most closely with the true data distribution at an intermediate layer (He et al., 2024). By employing the Wasserstein distance (Villani et al., 2008), this alignment is shown to reach a minimum at a specific depth—referred to as the *generalization funnel layer*. At this point, the *Min Wasserstein Generalization Bound* (He et al., 2024) ensures that the upper bound on the generalization gap—defined as the expected difference between the population and empirical risks—is minimized. This highlights the critical role of intermediate layers in supporting generalization.

**Research Question.** Despite the insight from prior studies of deep neural networks in classification, it remains unclear whether this generalizes to *Transformer-based language models in natural language generation (NLG)*. This motivates our central question: **In NLG, how does information**

**evolve across layers during training, and how are different layers of the network functionally organized to support generalization versus memorization?**

**Hypothesis.**  Building on the insight, we hypothesize that there exists a specific intermediate transformer layer that encodes the most generalizable representations for next-token prediction, characterized by maximal mutual information with the target label.

---

**Hypothesis: Generalization Ridge**

There exists an intermediate layer $\ell^* \in \{1, \ldots, L\}$ such that the mutual information between the hidden state and the target label peaks at that layer:

$$\ell^* = \arg\max_{\ell} I(Z_\ell; Y).$$

We refer to this layer as the ***generalization ridge***. This ridge layer aligns most strongly with generalizable features and serve as robust predictors under distribution shift, whereas later layers increasingly specialize in memorization.

---

**InfoRidge Overview.**  To empirically investigate this hypothesis, we propose InfoRidge, an information estimation framework, that characterizes how predictive information evolves across transformer layers. Specifically, we estimate two key quantities:

- ***Predictive Information*** $I(Z_\ell; Y)$: the mutual information between the hidden state at layer $\ell$ and the target token. This quantity measures how much information about the true next token is contained in the layer's full representation. A high value indicates that the layer encodes a strong and direct signal relevant to the prediction task.
- ***Incremental Information Gain*** $I(\Delta Z_\ell; Y)$: the information introduced by the residual transformation at layer $\ell$, where $\Delta Z_\ell = Z_\ell - Z_{\ell-1}$. This captures the additional predictive signal gained through the residual transformation at layer $\ell$, isolating how much new task-relevant information is introduced on top of the previous layer's representation.

Together, these metrics allow us to track both the accumulation and transformation of task-relevant information throughout the network.

**Notation and Setup.**  We consider a transformer model with $L$ residual blocks. Given an input sequence $x_{1:T}$ of length $T$, $z_i^{(\ell)} \in \mathbb{R}^d$ denotes the hidden state at the last token position of the $i$th input sequence in layer $\ell$, for $\ell = 1, \ldots, L$. For next-token prediction, the ground-truth label is denoted by $y_i \in \mathbb{R}^d$, corresponding to the embedding of the true next token from the vocabulary $\mathcal{V}$. The residual transformation introduced at layer $\ell$ is defined as $\delta z_i^{(\ell)} = z_i^{(\ell)} - z_i^{(\ell-1)}$.

Across a batch of $N$ sequences, we collect the representations:

$$Z_\ell = \{z_i^{(\ell)}\}_{i=1}^N, \quad Y = \{y_i\}_{i=1}^N, \quad \Delta Z_\ell = \{\delta z_i^{(\ell)}\}_{i=1}^N,$$

where all vectors are $\ell_2$-normalized.

**Computational Flow Overview.**  Figure 1 illustrates the workflow used to extract intermediate representations for information analysis. In the first forward pass, a batch of $N$ input sequences $x_{1:T}$ is fed into the transformer to obtain hidden states at each layer. From these, we extract the final-token representations $Z_\ell$ and compute the residual changes $\Delta Z_\ell$ by differencing consecutive layer outputs. In the second forward pass, each input is concatenated with its ground-truth next token $y$, and we extract the corresponding label embedding $Y$ from the output of the embedding layer. These representations are then used to compute two information-theoretic quantities: the *Predictive Information* $I(Z_\ell; Y)$, and the *Incremental Information Gain* $I(\Delta Z_\ell; Y)$. We estimate both $I(Z_\ell; Y)$ and $I(\Delta Z_\ell; Y)$ using Equation 1 and 2, detailed below.

**Matrix-Based Mutual Information.**  We apply the matrix-based framework (Giraldo et al., 2014) to estimate mutual information. Let $\mathcal{U}$ be a random variable, from which we draw a set of vectors

$U = \{\mathbf{u}_i\}_{i=1}^N \subset \mathbb{R}^d$. A positive-definite Gram matrix $G_U \in \mathbb{R}^{N \times N}$ is computed using a Gaussian kernel $\kappa$ with bandwidth set to 1 and the matrix is then trace-normalized to satisfy $\operatorname{tr}(G_U) = 1$. The matrix-based Rényi entropy (with order $\alpha = 1$) is then given by:

$$H(\mathcal{U}) \approx H(U) = -\operatorname{tr}(G_U \log G_U). \tag{1}$$

Specifically, $G_U$ is constructed with entries $(G_U)_{ij} = \exp\left(-\frac{\|u_i - u_j\|^2}{2\sigma^2}\right)$ and then trace-normalized.

Let $G_{\mathcal{U}}$ and $G_{\mathcal{V}}$ be the trace-normalized Gram matrices for two random variables $\mathcal{U}$ and $\mathcal{V}$, respectively. The mutual information between them is computed as:

$$I(\mathcal{U}; \mathcal{V}) \approx I(G_{\mathcal{U}}; G_{\mathcal{V}}) = H(G_{\mathcal{U}}) + H(G_{\mathcal{V}}) - H(G_{\mathcal{U}} \circ G_{\mathcal{V}}), \tag{2}$$

where "$\circ$" denotes the Hadamard (elementwise) product. Mathematical details are in Appendix A.

## 4 EXPERIMENTAL SETUP

**Models.** We evaluate four models: GPT–2 SMALL (117M) (Radford et al., 2019), GPT–2 MEDIUM (345M) (Radford et al., 2019), QWEN–2.5 0.5B (Yang et al., 2024), and LLAMA 3.1 8B (Meta AI, 2024). All models are fine-tuned on NLG tasks. Each model shares weights between the input token embedding and the output language modeling head.

**Datasets.** We assess model behavior across three tasks casted into NLG problems: CLUTRR (Sinha et al., 2019), a relational reasoning benchmark; ECQA (Aggarwal et al., 2021), a commonsense QA benchmark; and Synthetic Arithmetic, a controlled dataset designed to disentangle task-relevant signal from noise. Dataset and implementation details are in Appendix B and C.

*Synthetic Arithmetic Dataset Construction.* We construct a synthetic dataset to separate signal learning from noise memorization. Each input is a sequence of 10 elements, where the signal follows an arithmetic progression modulo $K$, computed as $s_t = (s_0 + t \cdot d) \bmod K$ with $s_0 \in [0, K-1]$ and $d \in [1, K-1]$. Each element takes the form S{signal}_N{noise}, where noise is sampled from $\mathcal{U}_{\text{int}}(0, \texttt{noise\_range}-1)$ ($\mathcal{U}_{\text{int}}$ denotes the uniform distribution). The model is trained to predict the signal value of the final (10th) element using the preceding elements as input context. For example, with $K = 5$, $s_0 = 1$, and $d = 2$, a sample input might be S1_N42 S3_N77...S2_N37, with the target signal being 4. By varying $K$, we can induce structured distribution shifts.

## 5 GENERALIZATION RIDGE: LAYER-WISE MUTUAL INFORMATION TRAINING DYNAMICS

Understanding how layers encode task-relevant information is key to uncovering the internal mechanisms that support generalization in deep language models. In this section, we trace the evolution of two complementary forms of mutual information—*Predictive Information* and *Incremental Information Gain*—across transformer depth and training time, revealing a consistent structure in information flow and highlighting the generalization–memorization trade-off.

### 5.1 PREDICTIVE INFORMATION: INFORMATION PEAKS AT INTERMEDIATE LAYERS

We investigate how *predictive information*—defined as the mutual information between hidden representations and target labels—evolves across the depth of transformer models. Specifically, for each layer $\ell$, we compute the matrix-based mutual information $I(Z_\ell; Y)$ between the hidden state $Z_\ell$ and the next-token ground truth $Y$. This quantity measures how much task-relevant signal is retained in the representation as it propagates through the network. By tracing $I(Z_\ell; Y)$ across layers, we obtain a layer-wise trajectory of information flow, which reveals not only where predictive content is preserved but also how it is transformed or diminished as the model processes.

Figure 2 tracks the trajectory of the predictive information $I(Z_\ell; Y)$ between the hidden representation at depth $\ell$ and the target label $Y$ throughout training, while Table 1 reports the downstream accuracy obtained when we early exit after a given layer.[1] Additional results are in Appendix E.

---

[1]The in distribution split is generated with $K_{\text{id}}=13$; out of distribution splits use a uniformly–sampled $K_{\text{ood}} \in \{5, \ldots, 25\} \setminus \{13\}$.

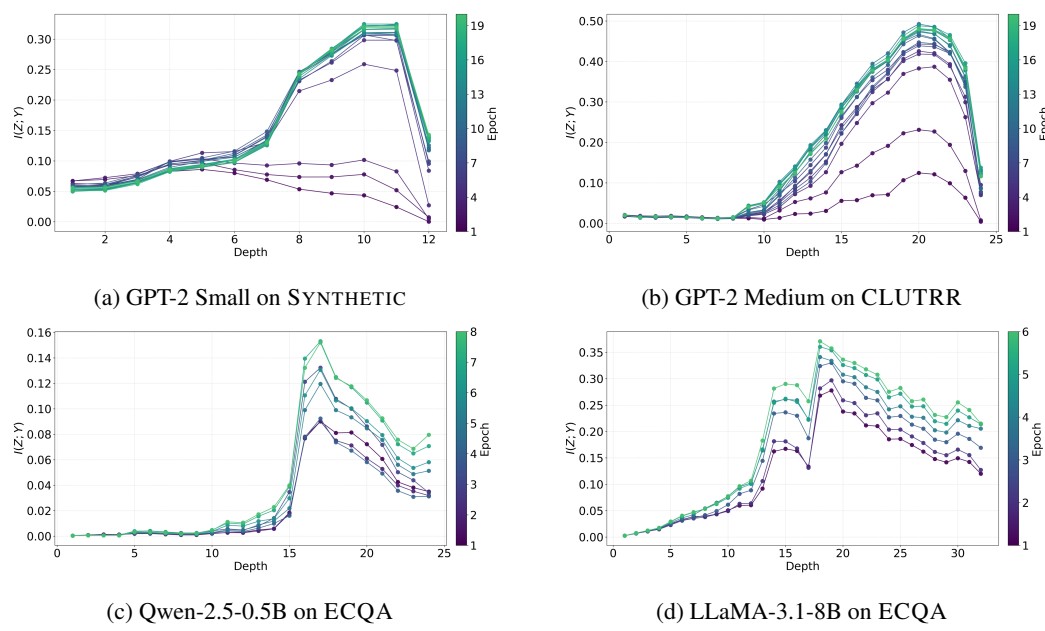

(a) GPT-2 Small on SYNTHETIC

(b) GPT-2 Medium on CLUTRR

(c) Qwen-2.5-0.5B on ECQA

(d) LLaMA-3.1-8B on ECQA

Figure 2: Evolution of predictive information $I(Z_\ell; Y)$, with lighter curves indicating later epochs. Each curve exhibits a three-phase trend: early layers rise, mid layers peak, and late layers decline.

**Three-phase information dynamics.** Across models and tasks, predictive information curves exhibit a consistent *three-phase* pattern:

*Progressive Accrual (early layers).* In the initial layers, $I(Z_\ell; Y)$ gradually increases, corresponding to basic feature extraction without substantial task-level comprehension. This aligns with the near-zero accuracy observed in Table 1 for these layers.

*Information Peak (intermediate layers).* $I(Z_\ell; Y)$ continues to rise through the mid-to-upper layer, typically peaking before the final few blocks. For the GPT-2 Small on Synthetic Arithmetic dataset, the peak reaches $I \approx 0.32$ in layers 10-11, coinciding with a jump from $\approx 0\%$ to $72\%$ ID accuracy and $53\%$ OOD accuracy. These layers appear to play a critical role in synthesizing abstract features that are essential for generalization.

*Representational Compression (final layers).* Beyond the peak, $I(Z_\ell; Y)$ decreases, even as in-distribution accuracy approaches $100\%$. The simultaneous drop in OOD accuracy indicates that the final layers tend to memorize training patterns, sacrificing generalization ability.

Table 1: Information Dynamics and Layer-wise Performance (GPT-2-S, Synthetic). OOD performance declines beyond the generalization ridge.

| Layer | $I(\mathbf{Z}; \mathbf{Y})$ | Test Accuracy (%) | | |
|---|---|---|---|---|
| | | **All** | **In-Dist.** | **Out-Dist.** |
| Layer 1 | 0.0508 | 0.00 | 0.00 | 0.00 |
| Layer 2 | 0.0513 | 0.00 | 0.00 | 0.00 |
| Layer 3 | 0.0619 | 0.00 | 0.00 | 0.00 |
| Layer 4 | 0.0822 | 0.00 | 0.00 | 0.00 |
| Layer 5 | 0.0894 | 0.00 | 0.00 | 0.00 |
| Layer 6 | 0.0989 | 0.00 | 0.00 | 0.00 |
| Layer 7 | 0.1292 | 0.70 | 0.00 | 2.00 |
| Layer 8 | **0.2431** | 18.35 | 0.00 | **38.90** |
| Layer 9 | 0.2810 | 19.15 | 1.00 | 40.30 |
| Layer 10 | **0.3209** | 62.45 | 71.90 | **53.50** |
| Layer 11 | **0.3209** | **72.65** | 99.90 | 40.40 |
| Layer 12 | **0.1402** | 71.45 | **100.00** | 38.60 |

Notably, this three-phase progression emerges consistently across both GPT-2 Small and GPT-2 Medium, indicating that the observed information dynamics are robust to architectural scale within the same model family.

**Generalization Ridge: Memorization-Generalization Trade-off.** The pronounced "information funnel" around intermediate layers reflects a key trade-off between generalization and memorization, which we term the "**generalization ridge**". These layers maximize task-relevant information for

generalization, while deeper layers increasingly compress and specialize representations, enhancing in-distribution memorization but reducing robustness. This positions intermediate layers as critical control points for managing this trade-off.

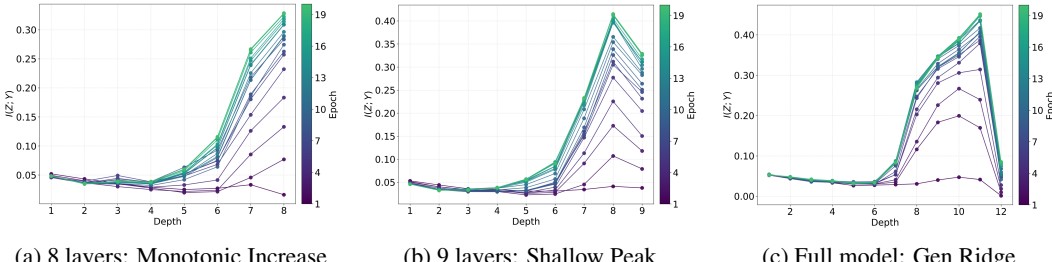

(a) 8 layers: Monotonic Increase  (b) 9 layers: Shallow Peak  (c) Full model: Gen Ridge

Figure 3: Truncating GPT–2 to 8 layers removes the MI peak; 9–layer variants begin to exhibit a shallow peak, and the full 12–layer model shows a pronounced decline.

**Generalization Ridge Emerges Beyond a Depth Threshold**   We empirically vary the depth of GPT-2 Small models fine-tuned on the CLUTRR dataset to examine how model capacity shapes information dynamics, shown in Figure 3. When the model is truncated to 8 layers, $I(Z;Y)$ increases monotonically—indicating insufficient capacity to develop an information peak. Upon increasing the depth to 9 layers, a shallow peak emerges, signifying the threshold at which the model begins to distinguish generalized features from memorized signals. The full 12-layer model exhibits a clear peak followed by a decline, confirming that the generalization ridge emerges only beyond a certain capacity threshold. These results underscore the role of architectural depth in information dynamics and sufficient capacity is required for the generalization ridge to emerge.

**Overfitting Adds Memorization in Final Layers**   To probe overfitting dynamics, we intentionally fine-tuned the model beyond the optimal point. In Figure 4, we observe that the Predictive Information $I(Z_\ell; Y)$ rises again in the final layers—departing from the typical compression phase expected at this stage. Before overfitting, the top layers largely behave as pretrained decoders, focusing on surface-level patterns such as token co-occurrence, syntactic templates, or corpus biases, which are not

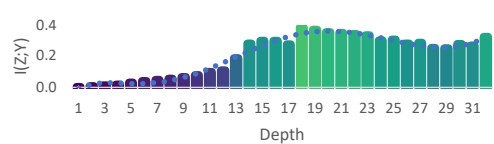

Figure 4: $I(Z_\ell; Y)$ rises in final layers during overfitting (LLaMA, ECQA).

strongly aligned with the task label—hence their initially low $I(Z_\ell; Y)$. After overfitting, however, the later-layer rise reflects a shift toward memorization, where the model begins to encode superficial shortcuts or redundant label-specific noise rather than useful task information.

**Semantically Important Attention Peaks Where Information Peaks**   This discussion aims to provide an intuitive illustration of our generalization ridge hypothesis, highlighting how attention to task-relevant signal tokens shifts across layers (the generalizable information). Specifically, we computed average signal attention across layers, identified the layer with peak signal attention, and compared it to final-layer signal and last-token attention, results are in Table 2. Signal tokens are defined as task-relevant tokens that the model must focus on to solve the task—for the synthetic dataset, these are tokens that appear after the character 'S'; for CLUTRR, kinship-related entities; and for ECQA, the token corresponding to the correct answer option. This allows us to quantify where in the network attention to semantically important information is concentrated. Our findings show a pattern that supports our hypothesis: (1) Mid-to-late layers peak in signal attention, coinciding with the predictive information ridge (Figure 2), indicating where generalizable representations are strongest. (2) Final layers show reduced attention to signal tokens, suggesting a shift toward memorization to specific data point rather than predictive information abstraction. For example, in Qwen-2.5-0.5B on ECQA, signal attention peaks at Layer 17 (0.2458) but drops to 0.0066 in the final layer, where last-token attention dominates (0.1697). Additional results are in Appendix F.

Table 2: Average attention statistics: (1) average attention scores over all tokens, (2) average attention to signal tokens, (3) the maximum signal attention and its corresponding layer, (4) signal token attention in the final layer and (5) last token attention in the final layer.

| Model | Dataset | Avg. Attn (All) | Avg. Attn (Signal) | Layer w/ Highest Avg. Signal | Final Avg. Signal | Final Avg. Last |
|---|---|---|---|---|---|---|
| GPT-2 Small | Synthetic | 0.0227 | 0.0483 | 10 (0.0758) | 0.0573 | 0.0400 |
| GPT-2 Medium | CLUTRR | 0.0086 | 0.0155 | 19 (0.0453) | 0.0147 | 0.0889 |
| Qwen-2.5-0.5B | ECQA | 0.0225 | 0.0257 | 17 (0.2458) | 0.0066 | 0.1697 |
| LLaMA-3.1-8B | ECQA | 0.0220 | 0.0379 | 21 (0.1276) | 0.0168 | 0.3904 |

*We remove the first token attention score to mitigate attention sink effects.

## 5.2 INCREMENTAL INFORMATION GAIN: INFORMATION CONCENTRATES AT INTERMEDIATE LAYERS

To understand how information accumulates across the network, we compute *Incremental Information Gain* ($I(\Delta Z_\ell; Y)$)—the mutual information between each residual transition and the target label embedding. As shown in Figure 5, the resulting layer-wise gains reveal that intermediate layers yield the highest information increases. This concentration of information gain further underscores their central role in encoding those task-relevant features that are essential for supporting generalization. For additional illustration, a detailed case study that decodes $\Delta z$ via the LM head is provided in Appendix G, highlighting fine-grained token-level shifts and variations observed across layers.

The Incremental Information Gain analysis reveals a clear pattern: middle transformer blocks are key to encoding generalizable task-relevant information, thereby forming a *generalization ridge*. Conversely, later layers contribute little additional predictive signal, and in some cases, actively reduce alignment with the target embeddings. This diminishing contribution in later layers may suggest a shift from general reasoning to memorization of training-specific patterns. This trend underscores a fundamental trade-off in transformer training dynamics.

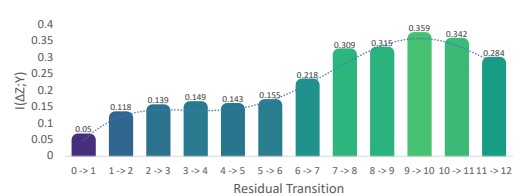

Figure 5: Middle blocks are key to encoding generalizable information (GPT-2-S, Synthetic).

## 6 RESIDUAL SCALING DYNAMICS VIA LEARNABLE $\beta$ COEFFICIENTS

To deepen our understanding of the Generalization Ridge hypothesis, we examine how modulating the contribution of individual transformer blocks affects information flow, and propose a corollary that links layer-wise contribution to generalization performance under distribution shift.

**Corollary** (Residual Scaling). *Post-peak residual blocks (i.e., layers beyond the generalization ridge) may encode memorized signals. Suppressing these layers' contribution via learned residual scaling improves out-of-distribution generalization, while amplifying them degrades it.*

We introduce a *residual scaling mechanism* with learnable scalar coefficient parameters (Algorithm 1), inspired by prior work on adaptive residual modulation (Liu et al., 2019a; Menghani et al., 2024). Transformer architectures inherently employ residual connections to iteratively refine representations. To isolate and quantify the contribution of each transformer block, we scale these residual connections with layer-specific scaling factors $\beta_\ell \in \mathbb{R}_{\geq 0}$, with definition below. Each $\beta_\ell$ controls the strength of the residual contribution from layer $\ell$, enabling the model to adaptively emphasize or suppress specific blocks:

$$z^{(\ell)} = z^{(\ell-1)} + \beta_\ell \cdot \text{block}^{(\ell)}(z^{(\ell-1)}), \qquad \beta_\ell \in \mathbb{R}_{\geq 0}.$$

**Definition** (Residual Scaling Coefficient). $\beta_\ell$ *is a learnable scalar parameter associated with transformer layer $\ell$, which modulates the contribution of that layer's residual output to the model's forward pass.*

We freeze model weights and optimize only the residual scaling coefficient parameters $\beta_\ell$, which are initialized to 1. These scalars are trained separately on the (a) in-distribution (ID) split and (b) out-

---

**Algorithm 1** Residual Scaling: Probing the Contribution of Transformer Blocks to the Generalization–Memorization Trade-off.

---

**Require:** Pretrained Transformer with $L$ layers, dataset $\mathcal{D}$, learning rate $\eta$
1: Initialize $\beta_1, \ldots, \beta_L \leftarrow 1.0$ (trainable); freeze other weights
2: **for** each training step **do**
3:     Sample batch $(x, y) \sim \mathcal{D}$
4:     $z^{(0)} \leftarrow \text{Embedding}(x)$
5:     **for** $\ell = 1$ to $L$ **do**
6:         $r^{(\ell)} \leftarrow \text{block}^{(\ell)}(z^{(\ell-1)})$
7:         $z^{(\ell)} \leftarrow z^{(\ell-1)} + \beta_\ell \cdot r^{(\ell)}$
8:     **end for**
9:     Compute loss $\mathcal{L}(z^{(L)}, y)$
10:     Update $\beta_1, \ldots, \beta_L$ using gradient descent
11: **end for**

---

of-distribution (OOD) split. Since no other parameters are updated, the learned $\beta_\ell$ serve as a direct diagnostic of the extent to which each layer should be amplified or attenuated to suit the data regime, revealing which layers remain stable across regimes and which adapt strongly to distributional shifts.

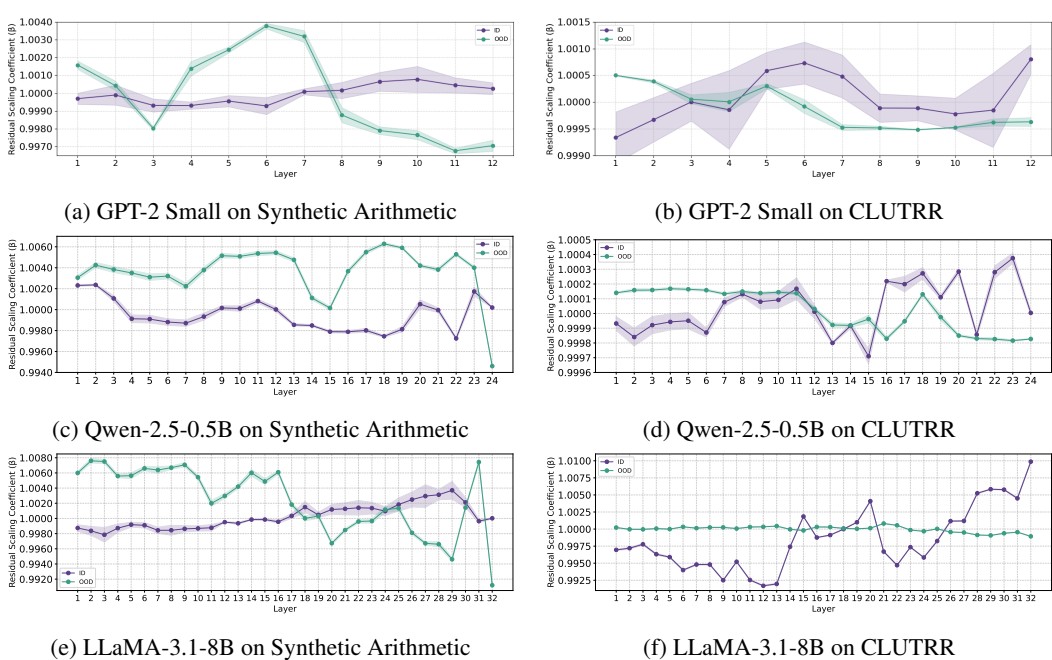

(a) GPT-2 Small on Synthetic Arithmetic      (b) GPT-2 Small on CLUTRR

(c) Qwen-2.5-0.5B on Synthetic Arithmetic      (d) Qwen-2.5-0.5B on CLUTRR

(e) LLaMA-3.1-8B on Synthetic Arithmetic      (f) LLaMA-3.1-8B on CLUTRR

Figure 6: Residual scaling coefficients $\beta_\ell$ across all transformer layers. ID training emphasizes later layers, while OOD training shifts weight toward middle layers, aligning with the generalization ridge. Curve shows the mean across 5 independent run, and the shaded region denotes 1-$\sigma$ error bar.

As shown in Figure 6, optimizing the residual scaling coefficient parameters on in-distribution data consistently yields $\beta_L > 1$, indicating that in-distribution performance benefits from amplifying the contribution of the final transformer layers. This observation suggests that these deeper layers specialize in memorizing features specific to the training distribution. In contrast, when trained on OOD settings, the learned coefficients exhibit $\beta_L < 1$, revealing that the model achieves better generalization by downweighting the influence of the final layers. Suppressing the contribution of these memorization-heavy components shifts the reliance back toward intermediate layers, which encode more generalizable and transferable signals. This pattern holds consistently across models, reinforcing the interpretation that model depth reflects a functional stratification. Intermediate layers concentrate generalizable information that supports generalization, whereas deeper layers become

increasingly specialized in memorized patterns tied to the training distribution. Together, these findings offer empirical evidence for the generalization ridge hypothesis, revealing that information flow in transformers reflects a trade-off between generalizable signals and memorized, distribution-specific features.

## 7 CONCLUSION

We introduce InfoRidge, an information-theoretic framework designed to trace and quantify how information evolves across layers in transformer-based language models for natural language generation. By estimating both predictive information and incremental information gain, we systematically characterize the layerwise dynamics of information flow, offering a principled view of how signals are refined, amplified, or diminished as they propagate through the network. Our findings reveal a consistent *generalization ridge* emerging in intermediate layers, where mutual information between the hidden representation and the target label reaches its peak before gradually declining. This phenomenon reflects a fundamental trade-off between generalization and memorization as information flows deeper into the model. Residual scaling experiments further corroborate this interpretation, demonstrating the functional specialization of different layers—where intermediate blocks play a key role in supporting generalization, while deeper layers increasingly focus on memorization. Taken together, these findings position InfoRidge as a comprehensive framework for diagnosing how language models internally manage information during natural language generation, while also revealing the structural mechanisms that govern the balance between generalization and memorization in transformer architectures.

**Ethics Statement.** This work adheres to the Code of Ethics. Our experiments use only open-source models and publicly available datasets under their respective open licenses, with no involvement of human subjects or sensitive data. We identify no foreseeable ethical risks.

**Reproducibility Statement.** We ensure reproducibility by providing experimental and implementation details in Section 4 and Appendices B–C. Full results with statistical significance are in Appendix E, and anonymous source code is included as supplementary material.

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

# A   MATHEMATICAL DETAILS FOR MATRIX-BASED INFORMATION ESTIMATION AND THEORETICAL FOUNDATIONS

We employ the matrix-based Rényi entropy Giraldo et al. (2014) to estimate mutual information between representations and labels, leveraging kernel Gram matrices to capture sample similarity.

## A.1   MATRIX-BASED ENTROPY ESTIMATION

Let $U = \{u_i\}_{i=1}^N \subset \mathbb{R}^d$ denote $\ell_2$-normalized representations obtained from a specific transformer layer, a residual update, or the embedding of the target label. A Gaussian kernel Gram matrix $G_U \in \mathbb{R}^{N \times N}$ is constructed as: $(G_U)_{ij} = \exp\left(-\frac{\|u_i - u_j\|^2}{2\sigma^2}\right)$, with bandwidth $\sigma = 1$. The matrix is then trace-normalized to ensure $\mathrm{tr}(G_U) = 1$.

The matrix-based Rényi entropy of order $\alpha = 1$ is defined as:

$$H(U) = -\mathrm{tr}(G_U \log G_U).$$

This expression can be interpreted in terms of the eigenvalue spectrum $\{\lambda_k\}$ of $G_U$, since $G_U$ is positive semi-definite and trace-normalized:

$$H(U) = -\sum_{k=1}^N \lambda_k \log \lambda_k.$$

The entropy thus reflects the dispersion of the eigenvalues. A more uniform spectrum (i.e., higher entropy) suggests more diversity in the representation space, while a sharply peaked spectrum (i.e., low entropy) indicates redundancy or compression.

## A.2   MUTUAL INFORMATION ESTIMATION

To estimate the mutual information between two random variables $U$ and $V$, we compute their Gram matrices $G_U$ and $G_V$, and form the joint similarity matrix via element-wise (Hadamard) product: $G_{UV} = G_U \circ G_V$. After trace-normalization, mutual information is estimated by: $I(U; V) = H(U) + H(V) - H(U, V)$, where $H(U, V) = -\mathrm{tr}(G_{UV} \log G_{UV})$. The eigenvalue spectrum of $G_{UV}$ governs the joint entropy term; its shape reflects how much of the structure in $U$ and $V$ aligns. A more concentrated spectrum in $G_{UV}$ relative to $G_U$ and $G_V$ implies stronger dependence and thus higher mutual information.

## A.3   MIN WASSERSTEIN GENERALIZATION BOUND (HE ET AL., 2024)

We restate the generalization bound proposed by He et al. (He et al., 2024), which characterizes generalization in terms of the Wasserstein distance between internal representations.

Suppose that the loss function $\tilde{\ell} : \mathcal{Y} \times \mathcal{Y} \to \mathbb{R}_{\geq 0}$ is $\rho_0$-Lipschitz, and the activation function $\phi_\ell : \mathbb{R} \to \mathbb{R}$ is $\rho_\ell$-Lipschitz for each $\ell = 1, \ldots, L$. Then:

$$\mathrm{gen}(P_{W|\mathcal{D}_n}, P_{X,Y}) \leq \min_\ell \frac{\rho_0}{n} \sum_{i=1}^n \mathbb{E}_W \left[ \left(1 \vee \prod_{j=\ell+1}^L \rho_j \|W_j\|_{\mathrm{op}}\right) \cdot \mathcal{W}_1\left(P_{T_\ell, i, Y_i|W}(\cdot|W), P_{T_\ell, Y|W}(\cdot|W)\right) \right] \quad (3)$$

This result shows that generalization can be tightly controlled by the Wasserstein distance (Villani et al., 2008) between representations at a specific layer—referred to the generalization funnel layer.

**Connection to Our Work.**   The *Min Wasserstein Generalization Bound* (He et al., 2024) provides a theoretical foundation for our study by characterizing generalization in terms of distributional alignment at an intermediate layer. It motivates our analysis of information flow by suggesting that information is peaked at a specific layer—the generalization funnel. Our InfoRidge builds on this insight by quantifying predictive information across layers, and reveals that a specific intermediate layer exhibits peak mutual information and correlates with better generalization performance.

## B    DATASET OVERVIEW AND STATISTICS

To evaluate information flow and generalization dynamics across model layers, we conduct experiments on three datasets with varying levels of complexity and structure: CLUTRR, ECQA, and a custom-designed Synthetic Arithmetic dataset. Table B.1 summarizes key dataset statistics.

### B.1    DATASET OVERVIEW

Table 3: Dataset Statistics

| Dataset | #Train | Train Seq. Len | #Val | Val Seq. Len | #Test | Test Seq. Len |
|---|---|---|---|---|---|---|
| **CLUTRR** | 9,074 | 30 | 2,020 | 29 | 1,146 | 70 |
| **ECQA** | 7,598 | 21 | 1,090 | 21 | 2,194 | 21 |
| **Synthetic Arithmetic** | 10,000 | 9 | 2,000 | 9 | 2,000 | 9 |

### B.2    CLUTRR

CLUTRR (Compositional Language Understanding and Text-based Relational Reasoning) (Sinha et al., 2019) is a diagnostic benchmark for evaluating relational inference in language models. Each example contains a story describing family relations, and the task is to infer the missing relationship between two entities. The distribution shift stems from clause lengths that are absent in the training set but present during evaluation. We use the task split "gen_train23_test2to10", where the model is trained on clause lengths 2 and 3 and evaluated on lengths 2 through 10.

### B.3    ECQA

ECQA (Explanations for CommonsenseQA) (Aggarwal et al., 2021) is a commonsense multiple-choice question-answering dataset, where each question is accompanied by 5 answer options.

### B.4    SYNTHETIC ARITHMETIC DATASET

We construct a synthetic diagnostic dataset to disentangle task-relevant signal learning from spurious noise memorization in a controlled setting. Each sample consists of a sequence of 10 symbolic elements, where the signal component follows an arithmetic progression modulo $K$, and the remainder of each element is independently corrupted with random noise. By varying the modulus $K$, we systematically control task complexity and introduce structured shifts in the data distribution.

**Synthetic Arithmetic Dataset Construction.**    At each position $t$, the signal value is computed as:

$$s_t = (s_0 + t \cdot d) \bmod K, \quad \text{with } s_0 \in [0, K{-}1], \quad d \in [1, K{-}1].$$

Each element in the sequence is represented as a string of the form:

$$\texttt{S\{signal\}\_N\{noise\}}, \quad \text{where noise} \sim \mathcal{U}_{\text{int}}(0, \texttt{noise\_range-1}).$$

Here $\mathcal{U}_{\text{int}}$ denotes the uniform distribution. The model is trained to predict the signal value of the final (10th) element, using the preceding elements as input context.

For example, with $K = 5$, $s_0 = 1$, and $d = 2$, a sample might look like:

```
S1_N42 S3_N77 S0_N18 S2_N56 S4_N90 S1_N11 S3_N65 S0_N23 S2_N37
```

Each element encodes both a signal (the number following `S`) and a noise component (the number following `N`). The target is `4`, corresponding to the signal of the final (10th) item in the sequence.

## C    IMPLEMENTATION DETAILS

This appendix outlines implementation details in our experiments.

## C.1 PROMPT CONSTRUCTION

All tasks are cast into a next-token generation format. The model receives a prompt and generate the next token. Below are construction strategies and examples for each dataset:

**CLUTRR**  Each input example in CLUTRR consists of a short narrative describing a set of family relationships, along with a query involving a pair of entities. We construct prompts by concatenating the narrative and a structured natural language question derived from the query tuple. The model is trained to predict the correct relationship as the next token.

**Prompt:**

> Story: [Alice] is [Bob]'s mother. [Bob] is [Charlie]'s father.
> Query: What is the relationship between Alice and Charlie? Answer:

**Target:** grandmother

**ECQA (Explanation-augmented Commonsense QA)**  Each ECQA instance consists of a multiple-choice question with five candidate answers. We format the prompt by presenting the question followed by all five options (labeled A–E), and conclude with an explicit answer query. The model is trained to predict the correct answer letter as the next token.

**Prompt:**

> Question: What do people usually do at a birthday party?
> Options:
> A. Sleep
> B. Celebrate
> C. Cook
> D. Exercise
> E. Drive
> Answer:

**Target:** B

**Synthetic Arithmetic**  Each synthetic sample consists of a sequence of 10 symbolic elements, where each element is formatted as `S{signal}_N{noise}`. The signal values follow an arithmetic progression modulo $K$, and the noise values are independently sampled from a uniform distribution with a fixed range of 100. During training, the modulus $K$ is set to 13. For evaluation, test sequences are generated using values of $K$ from the range $[5, 26]$ excluding 13 to simulate a distribution shift. In the residual $\beta_\ell$ analysis, we use $K = 13$ for in-distribution (ID) training and $K = 17$ for out-of-distribution (OOD) training, allowing for a controlled comparison between generalization and memorization behavior. The model receives the first 9 tokens as input and is trained to predict the signal component of the 10th token.

**Prompt:**

> S1_N42 S3_N88 S5_N20 S7_N10 S9_N65 S11_N43 S0_N99 S2_N38 S4_N77

**Target:** 6

This controlled format enables manipulation of distributional properties by varying the modulus $K$.

## C.2 FINETUNING SETTINGS

We fine-tuned all layers end-to-end using AdamW (learning rate $5 \times 10^{-6}$, weight decay 0.01) with a linear schedule (warmup ratio 0.1) and early stopping on validation loss. Training converged in all settings.

## C.3 INFORMATION ESTIMATION

To estimate mutual information, we subsample between 50 and 200 test examples depending on model and task to achieve a stable result.

## C.4 Residual Scaling with Learnable $\beta_\ell$ Parameters

We introduce a vector of learnable scalar weights $\beta = \{\beta_1, \ldots, \beta_L\}$ applied to residual connections in a frozen transformer:

$$z^{(\ell)} = z^{(\ell-1)} + \beta_\ell \cdot \text{Block}^{(\ell)}(z^{(\ell-1)}).$$

- All transformer weights are frozen; only $\beta_\ell$ parameters are trained.

- Each transformer block's residual output is modulated by $\beta_\ell$ via forward hooks.

- $\beta$ are initialized to 1 and updated using gradient descent without altering the architecture.

**Training and Evaluation Settings** We perform residual $\beta_\ell$ training on top of a model that has already been fine-tuned on the target dataset, keeping all previously learned weights frozen and optimizing only the $\beta_\ell$ parameters using HuggingFace's Trainer API. We sweep over batch sizes from 4 to 32 depending on GPU memory constraints, and tune the learning rate between $5 \times 10^{-5}$ and $5 \times 10^{-6}$. We train for 3–10 epochs based on model capacity and task complexity. Optimization is performed using the AdamW optimizer with a weight decay of 0.01. Evaluation is conducted by comparing the next predicted token with the next ground-truth target token.

Table 4: Performance (%) of different models across datasets.

| Model | CLUTRR | Synthetic | ECQA |
|---|---|---|---|
| GPT-2 Small | 30.98 | 71.45 | — |
| GPT-2 Medium | 34.21 | 68.35 | — |
| Qwen-2.5-0.5B | 31.94 | 83.20 | 57.06 |
| LLaMA-3.1-8B | 57.68 | 94.00 | 77.39 |

## D Compute and Licensing Details

**Computing Resources** Experiments were conducted on an NVIDIA RTX A6000 GPU (48GB).

**Model Licenses** The GPT-2 Small (Radford et al., 2019) and GPT-2 Medium (Radford et al., 2019) models are released under the MIT License and are available via Hugging Face Transformers. The Qwen-2.5 0.5B (Yang et al., 2024) model is provided by Alibaba Group under the Apache 2.0 License. The LLaMA-3.1 8B (Meta AI, 2024) model is made available by Meta under a non-commercial research license and accessed via Hugging Face.

**Dataset Licenses** CLUTRR (Sinha et al., 2019) is released under a CC BY 4.0 license as part of EMNLP 2019. The ECQA (Aggarwal et al., 2021) dataset is also released under a CC BY 4.0 license by its authors as part of EMNLP 2021. The Synthetic Arithmetic dataset is custom-designed by the authors and does not rely on any external or licensed data sources.

All assets were used in compliance with their respective licenses, and no proprietary or restricted resources were employed in our experiments.

## E Full Quantitative Results with Confidence Estimates

To provide a comprehensive view of model behavior, we report the full experimental results for all models and datasets. This includes predictive information trends ($I(Z; Y)$), incremental information gain ($I(\Delta Z; Y)$) and residual scaling effects ($\beta_\ell$), with statistical significance reporting.

### E.1 Predictive Information

To estimate mutual information between hidden representations and target labels, we compute matrix-based mutual information following the methodology in Appendix A. Due to the high memory cost of computing $N \times N$ Gram matrices, we subsample between 50 and 100 test examples per experiment and report results with statistical significance based on multiple random seeds.

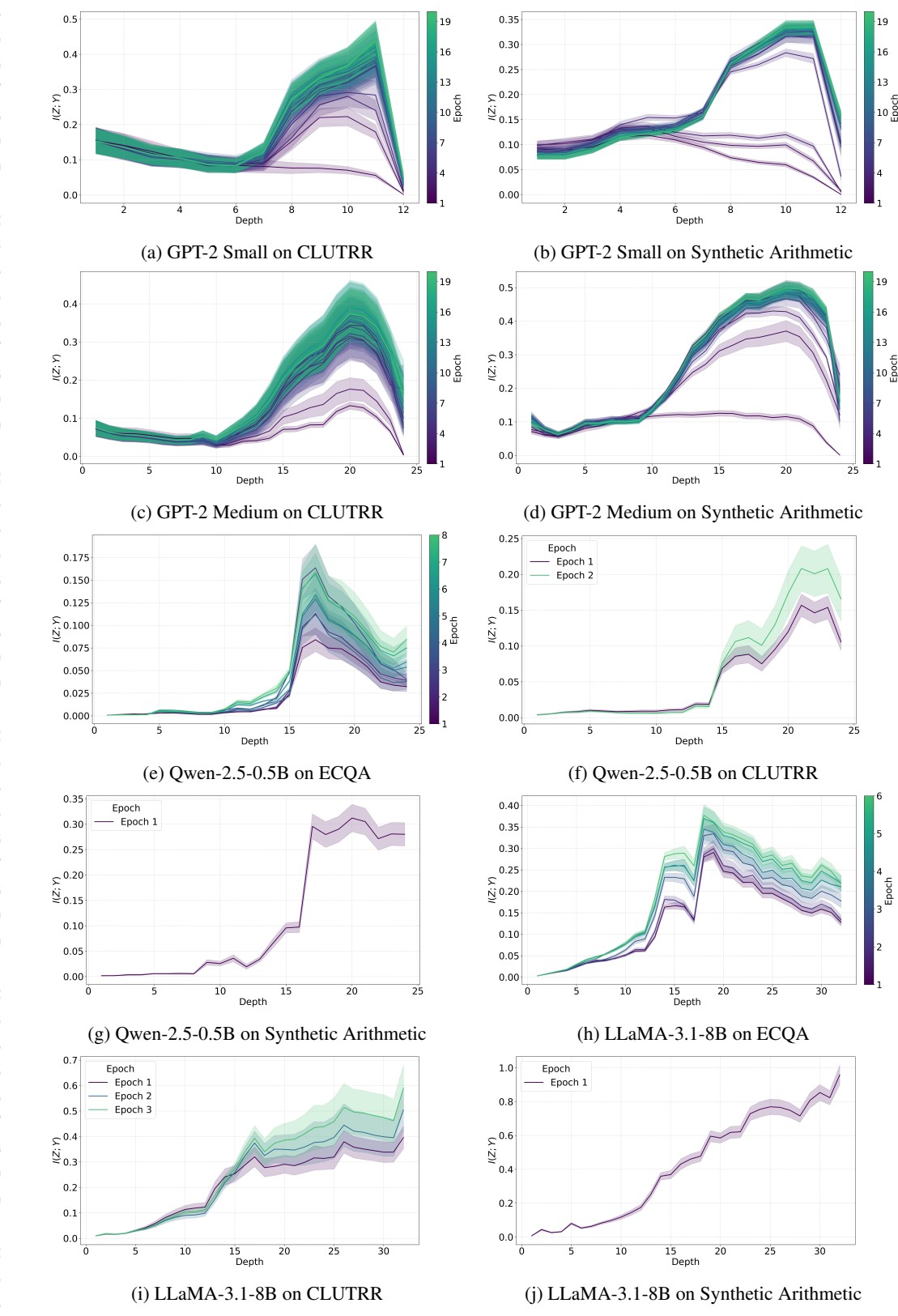

Figure 7: Predictive information $I(Z; Y)$ across different models and datasets exhibits an information peak, indicating a generalization ridge. In cases where the task is too simple relative to model capacity—such as the synthetic arithmetic task with LLaMA—this trend reflects an overfitting regime. Lighter line colors represent later training epochs. Each curve shows the mean across 5 random seeds (0, 1, 2, 3, 42), and the shaded region denotes a 2-sigma (~96%) confidence interval.

## E.2 Incremental Information Gain

In addition to predictive information $I(Z^{(\ell)}; Y)$, we compute the incremental information gain $I(\Delta Z^{(\ell)}; Y)$ at each layer. This quantifies the information contribution for each transformer block.

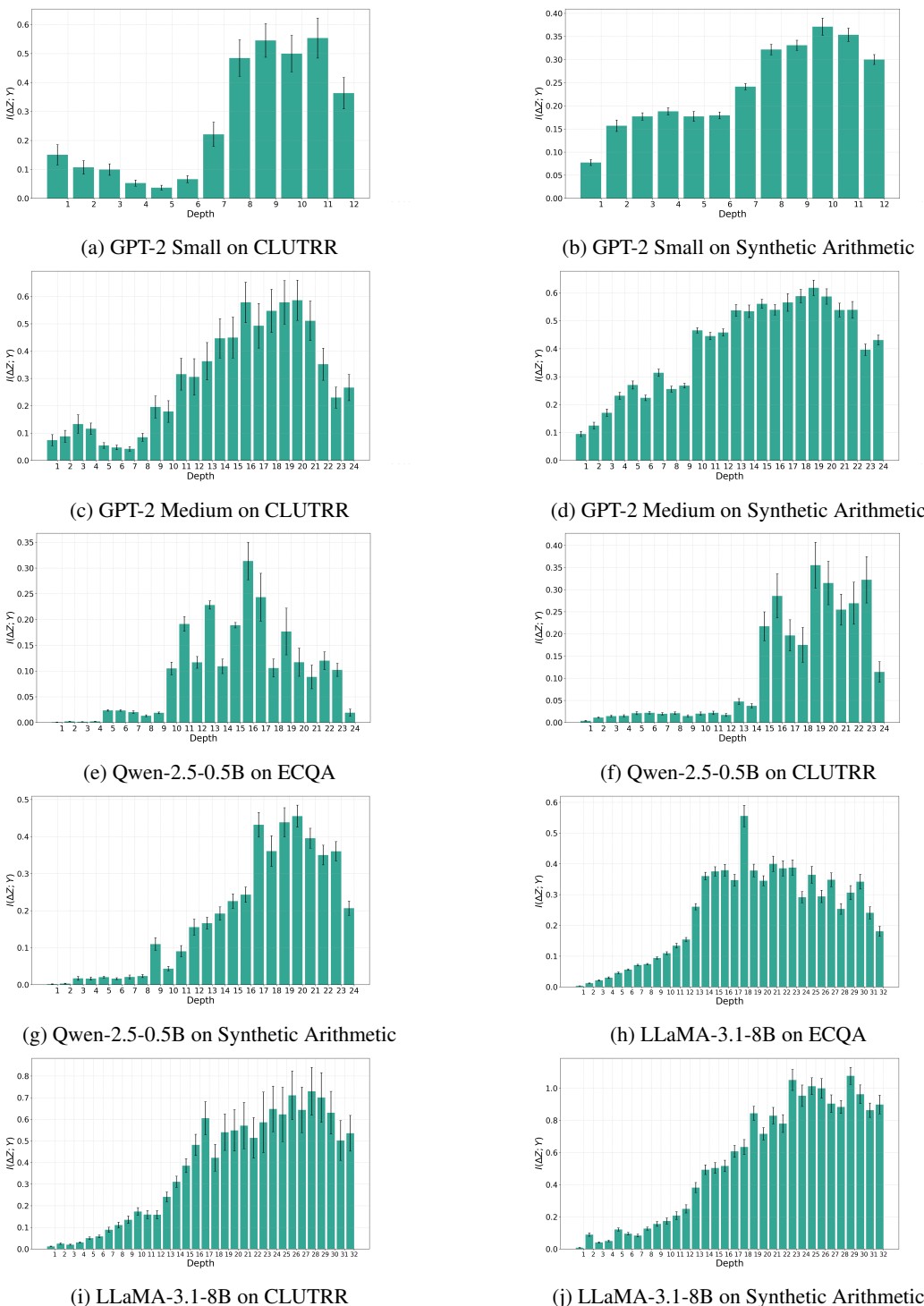

Figure 8: Incremental information gain $I(\Delta Z; Y)$ across different models and datasets with ~96% CI error bars. Across all models, we observe that the largest incremental information gain consistently occurs in intermediate layers—further supporting the emergence of a generalization ridge.

### E.3 RESIDUAL SCALING

We introduce a *residual scaling mechanism* with learnable scalar coefficient parameters, inspired by prior work on adaptive residual modulation (Liu et al., 2019a; Menghani et al., 2024). Similar ideas of modulating internal computation have also been explored in parameter-efficient fine-tuning (PEFT) (Houlsby et al., 2019), Representation-Efficient Fine-Tuning (REFT) (Wu et al., 2024) and interpretability-driven control (Huang et al., 2024; Deng et al., 2025; Meng et al., 2022; Wu et al., 2023). We present the complete set of residual scaling results, detailing the learned $\beta_\ell$ values across all transformer layers. These values reflect the relative contribution of each layer after optimizing the residual scaling coefficients while keeping all other model parameters frozen. $\beta_\ell$ are trained on in-distribution (ID) and out-of-distribution (OOD) data. We observe that in the ID setting, later layers tend to receive higher weights, consistent with memorization behavior. In contrast, OOD training consistently downweights final layers and shifts importance toward the middle of the network, aligning with the generalization ridge identified through InfoRidge.

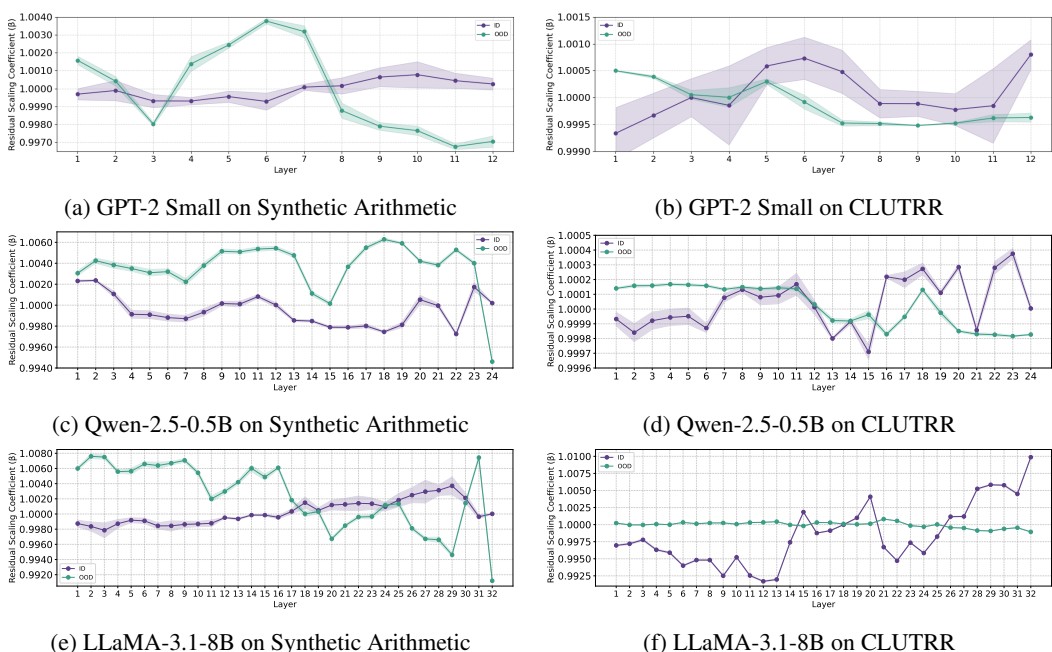

(a) GPT-2 Small on Synthetic Arithmetic      (b) GPT-2 Small on CLUTRR

(c) Qwen-2.5-0.5B on Synthetic Arithmetic      (d) Qwen-2.5-0.5B on CLUTRR

(e) LLaMA-3.1-8B on Synthetic Arithmetic      (f) LLaMA-3.1-8B on CLUTRR

Figure 9: Residual scaling coefficients $\beta_\ell$ across all transformer layers. ID training emphasizes later layers, while OOD training shifts weight toward middle layers, aligning with the generalization ridge observed via InfoRidge. Each curve shows the mean across five random seeds (0, 1, 2, 3, 42), and the shaded region denotes 1-sigma error bar.

## F ATTENTION DYNAMICS ACROSS LAYERS CASE STUDY

To verify whether generalization ridge layer indeed correspond to semantically meaningful processing, we visualize the attention patterns as a more interpretable signal of where the model focuses. As shown in Figure 10, we visualize attention maps across layers.

In early layers (e.g., Layer 1), attention is diffuse and biased toward final tokens—reflecting reliance on position rather than true predict signals.

By generalization ridge layers (e.g., Layer 11), attention becomes more targeted, concentrating on predictive tokens. This shift marks a transition from positional attention to semantically meaningful focus, suggesting that intermediate layers are increasingly capable of isolating task-relevant information from irrelevant content.

In the final layers, attention regresses toward final tokens. This reversion is aligned with the observed decline in $I(Z_\ell; Y)$. The resurgence of attention towards terminal tokens indicates a potential memo-

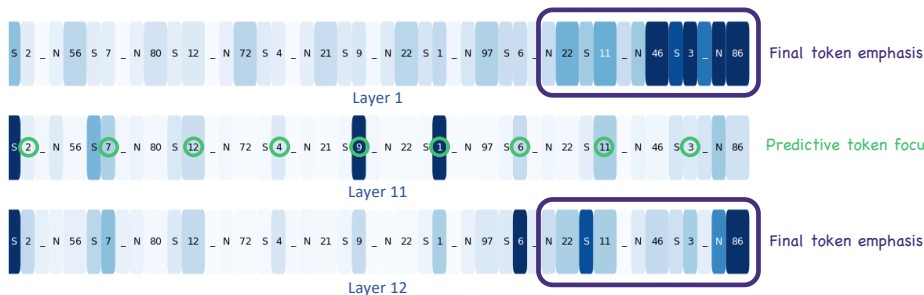

Figure 10: Attention map across layers (GPT-2 Small, Synthetic). At the generalization ridge layers, attention becomes more targeted, focusing on predictive tokens.

rization scenario, where the model re-engages superficial positional strategies, possibly memorizing noise rather than further refining the generalized extraction of predictive signals.

This attention trajectory further supports the generalization ridge hypothesis, highlighting a trade-off between generalization and memorization in the model's representational strategy.

## G INCREMENTAL INFORMATION GAIN CASE STUDY

To further illustrate how intermediate layers contribute to generalization, we analyze the semantic content introduced by residual transitions at different depths. Specifically, we decode the residual transition $\delta z$ at each layer using the language modeling (LM) head, projecting the incremental representation back into token space.

This analysis allows us to inspect the linguistic shift introduced by each transformer block in isolation, and to assess whether the changes correspond to task-relevant predictions or superficial noises.

### G.1 METHODOLOGY

For a fixed input, we compute residual transitions $\delta z^{(\ell)}$ at each layer and pass them through the model's final linear projection (LM head) followed by a softmax. We record the top predicted tokens and their probabilities, and it reflects the directional change applied by layer $\ell$.

### G.2 OBSERVATIONS

We decode residual transitions across layers and report the top shifted tokens by projecting $\Delta Z^{(\ell)}$ through the LM head. These shifts provide insight into how each layer modifies the model's internal prediction trajectory.

Layer $9 \rightarrow 10$: The top shifted tokens include `GNine`, `G10`, `G4`, `G8`, and `8`, which are all numerically aligned with the target prediction space in the synthetic arithmetic task. This indicates that the model is beginning to refine task-relevant numerical features at this depth.

Layer $10 \rightarrow 11$: The shifted tokens become partially diluted, featuring punctuation and less informative symbols such as `,`, `G`, and `.`, alongside occasional task-relevant entries like `G12` and `G4`. This indicates a transitional phase where the model continues to refine meaningful task-relevant features, yet begins to exhibit noise from frequent but semantically uninformative, such as punctuation, tokens.

Layer $11 \rightarrow 12$: At the final layer transition, the top 5 shifted tokens become largely uninterpretable, including `Gcanaver`, `soDeliveryDate`, `enegger`, `76561`, and `ILA`. Conversely, the most negatively shifted tokens—`G4`, `G3`, `G5`, `G6`, `G1`—correspond to plausible numerical predictions that were actively suppressed. This supports the hypothesis that final layers may overwrite generalizable abstractions with memorized or noise signals.

These patterns are consistent with our mutual information analysis, which identifies intermediate layers as better semantically aligned with the prediction target—forming a ridge of generalization.

### G.3 IMPLICATION

These decoding results reinforce our interpretation of the generalization ridge: intermediate layers contribute the most semantically informative updates to the model's representation. The residual transitions thus serve as a useful lens for understanding how and where semantic meaning is introduced during forward propagation.

## H ACTIVATION PATCHING EXPERIMENT

To assess which residual blocks causally support OOD prediction, we conduct a targeted activation-patching experiment. Let the model contain $L$ residual blocks with hidden states $Z_\ell \in \mathbb{R}^d$, and let $t^*$ denote the final context token (the last token before next-token prediction). For every OOD sequence $x_{\text{ood}}$, we pair it with an ID sequence $x_{\text{id}}$ sharing the same token structure. For each depth $\ell \in \{1, \ldots, L\}$, we compute two forward passes: a standard OOD pass and a patched pass in which we replace only the hidden state of token $t^*$ at depth $\ell$ according to $Z_\ell(x_{\text{ood}}, t^*) \leftarrow Z_\ell(x_{\text{id}}, t^*)$. All other activations remain unchanged. Let $logit_{\text{base}}$ denote the logit assigned to the correct next token under the baseline OOD pass, and let $logit_{\text{patch}}^{(\ell)}$ denote the corresponding logit under layer-$\ell$ patching. We measure causal impact using the logit drop $\Delta_\ell = logit_{\text{base}} - logit_{\text{patch}}^{(\ell)}$, where a positive value indicates that depth $\ell$ provides information necessary for correct OOD prediction.

Table 5 reports the averaged logit drop for all $\ell = 1, \ldots, 12$ (GPT-2 Small on Synthetic Dataset). The effects are negligible for layers 1–8, begin rising around layer 9, increase sharply at layer 10, peak at layer 11, and decline again at layer 12. This rise–peak–fall pattern closely matches the ridge structure observed in Figure 2, where $I(Z_\ell; Y)$ increases through mid-to-late depth, reaches a maximum in the ridge region, and then decreases. The activation-patching experiment therefore provides supporting causal evidence that the ridge layers identified by IndoRidge are necessary for successful OOD generalization.

Table 5: Average logit drop $\Delta_\ell$ (baseline logit - patched logit). Larger values indicate greater causal contribution to OOD prediction.

| Layer $\ell$ | 1 | 2 | 3 | 4 | 5 | 6 | 7 | 8 | 9 | 10 | 11 | 12 |
|---|---|---|---|---|---|---|---|---|---|---|---|---|
| $\Delta_\ell$ | -0.1037 | -0.1039 | -0.0327 | -0.5405 | -5.1663 | -8.3992 | -5.8914 | -15.0996 | -3.0998 | **9.3148** | **17.2022** | 15.5943 |

## I EXTENDING PREDICTIVE INFORMATION TO MULTI-TOKEN OUTPUTS

To complement our single-token analysis, we further evaluate information flow under multi-token outputs using the CNN/Daily Mail summarization dataset (1.0.0, test) (See et al., 2017; Hermann et al., 2015).

**Autoregressive next-token information.** We follow the model's natural decoding process. We analyze $I(Z_\ell^{(t)}; \hat{Y}^{(t+1)})$ since there is no deterministic gold answer for each intermediate step in NLG, where for each generation step $t$, $Z_\ell^{(t)}$ is the hidden state of current token at layer $\ell$ and $\hat{Y}^{(t+1)}$ is the embedding of model's predicted token at step $t + 1$.

Across steps, the pattern is consistent (Figure 11): information rises from early layers, peaks in the middle layers, and then declines toward top layers. This gives a clear information ridge, indicating that intermediate layers encode the most stable and task-relevant core information that supports the model's decision-making (generalization). In particular, high $I(Z_\ell^{(t)}; \hat{Y}^{(t+1)})$ at these layers means the model's predictive direction is most strongly determined by that layer: layers whose representations consistently constrain the next-token prediction across many contexts must encode the shared, invariant structure that the model relies on to generalize. In contrast, upper layers exhibit behavior that might be related to pretrained surface patterns from corpus such as token co-occurrence, causing the representation to drift away from this decision-relevant structure (memorization).

The first step is a notable outlier: it is the only step that conditions solely on the input prompt, and shows the sharpest and most pronounced ridge. Generating the first token requires a global transition

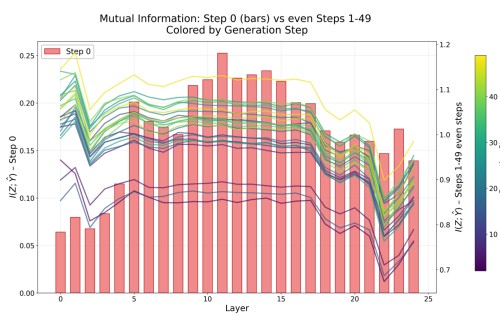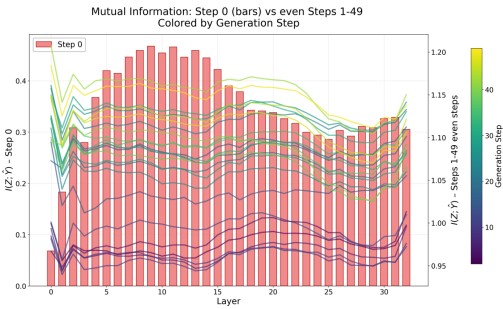

(a) Qwen-2.5-0.5B on CNN/Daily Mail for 50 generation steps.

(b) LLaMA-3.1-8B on CNN/Daily Mail for 50 generation steps.

Figure 11: $I(Z_\ell^{(t)}; \hat{Y}^{(t+1)})$ across layers (x-axis) and generation steps (y-axis), where the left y-axis shows first step mutual information (bars) and the right y-axis shows mutual information for the rest steps.

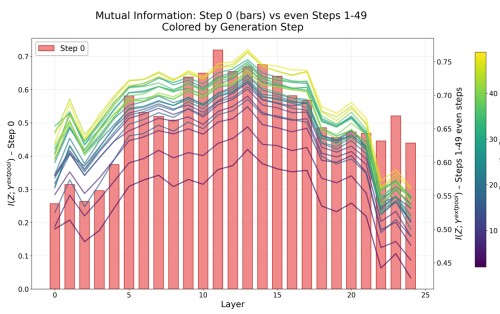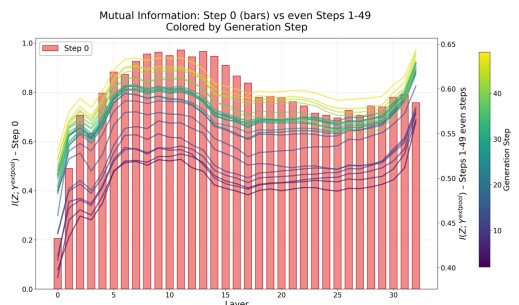

(a) Qwen-2.5-0.5B on CNN/Daily Mail for 50 generation steps.

(b) LLaMA-3.1-8B on CNN/Daily Mail for 50 generation steps.

Figure 12: $I(Z_\ell^{(t)}; Y^{\text{pool}})$ across layers (x-axis) and generation steps (y-axis), where the left y-axis shows first step mutual information (bars) and the right y-axis shows mutual information for the rest steps.

from the prompt into the answer trajectory, which heavily engages the abstract, decision-relevant subspace in the middle layers. Crucially, this is also the point where information compression is most extreme, the model must condense all prompt-level semantics into a single initial token decision, therefore forms a substantial ridge. Once generation begins, later generation steps move into a more regular autoregressive regime, where each hidden state already contains a compressed summary of all previously generated content, and the continuation depends more on local context than on global restructuring. Consequently, the depth profiles become flatter and more stable across steps, although the mid-layer ridge remains visible.

**Pooled-answer information.** We quantify how much information each layer carries about the gold answer by computing $I(Z_\ell^{(t)}; Y^{\text{pool}})$, where $Y^{\text{pool}}$ is a fixed target vector obtained by average-pooling the embedding-layer representations of all ground-truth answer tokens. Across models and steps (Figure 12), a consistent pattern emerges: intermediate layers are where the hidden state is most strongly aligned with the gold-answer.

Taken together, the two constructions reveal a coherent picture of depth specialization in multi-token generation. Intermediate layers are simultaneously the most informative about the model's next-token decisions and the most aligned with the global answer. Thus, the two observed ridges are complementary projections of a single underlying phenomenon: intermediate layers concentrated most meaningful and decision-relevant information. In contrast, upper layers increasingly reflect pretrained corpus-specific surface patterns, causing the representation to drift away from this decision-relevant structure.

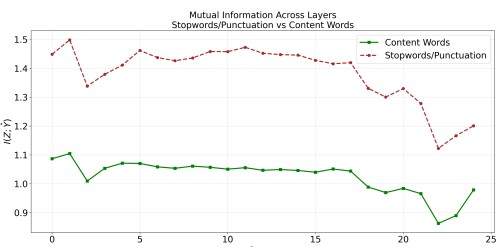 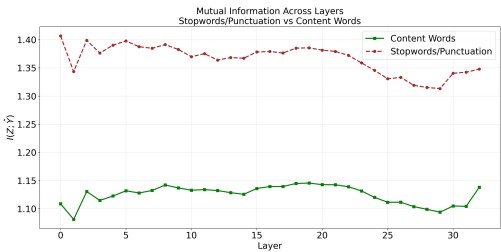

(a) $I(Z_\ell^{(t)}; \hat{Y}^{(t+1)})$: Qwen-2.5-0.5B on CNN/Daily Mail for 50 generation steps.

(b) $I(Z_\ell^{(t)}; \hat{Y}^{(t+1)})$: LLaMA-3.1-8B on CNN/Daily Mail for 50 generation steps.

Figure 13: Mutual information comparing stopwords/punctuation against content words, showing that both stopwords/punctuation have consistently higher MI.

We also plot the mutual information comparing stopwords/punctuation against content words, showing that both stopwords/punctuation have consistently higher MI (Figure 13). These tokens are more predictable because they follow strict grammatical and structural patterns, whereas content words depend more on semantic context and therefore exhibit greater variability.

## J    NON-FINETUNED SCENARIO

Beyond fine-tuning scenarios, we also computed predictive-information $I(Z_\ell; Y)$ curves in a non-training zero-shot regime and observed the same mid-layer information peak pattern, see Figure 14

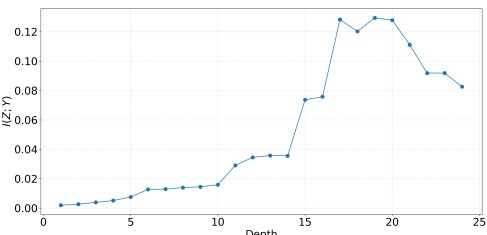

Figure 14: GSM8K on Qwen2.5-0.5B shows an intermediate-layer MI rise and late-layer decline.

## K    KERNEL ABLATION

The matrix-based MI estimator requires a positive-definite Gram matrix, for which the Gaussian kernel is the standard choice. To assess robustness with respect to kernel selection, we additionally test the Laplacian and Polynomial kernels. While these kernels introduce quantitative shifts in the magnitude, the trend—including the rise–peak–decline structure and the location of the ridge—remains unchanged (Figure 15).

**Polynomial Kernel.**    Given a set of normalized representations $U = \{u_i\}_{i=1}^N$, the polynomial kernel Gram matrix is defined as

$$(G_U^{\text{poly}})_{ij} = \left( u_i^\top u_j + c_0 \right)^p, \tag{4}$$

where $p$ is the polynomial degree and $c_0$ is a constant bias term.

**Laplacian Kernel.**    Given the same representation set $U = \{u_i\}_{i=1}^N$, the Laplacian kernel Gram matrix is defined as

$$(G_U^{\text{lap}})_{ij} = \exp(-\gamma \|u_i - u_j\|_1), \tag{5}$$

where $\gamma > 0$ is a kernel coefficient and $\|\cdot\|_1$ denotes the $\ell_1$ distance.

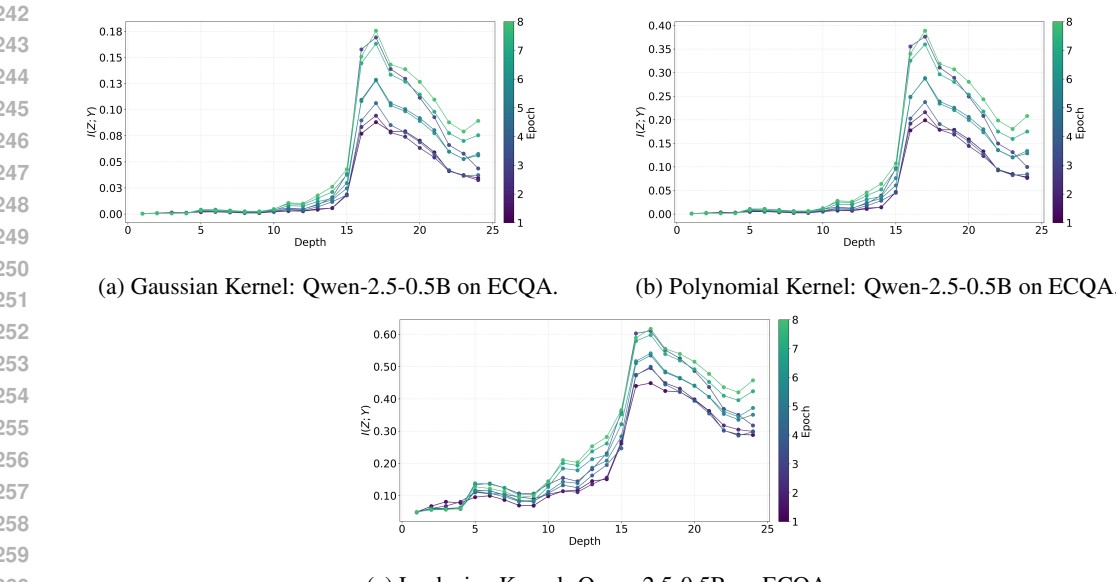

(a) Gaussian Kernel: Qwen-2.5-0.5B on ECQA.  (b) Polynomial Kernel: Qwen-2.5-0.5B on ECQA.

(c) Laplacian Kernel: Qwen-2.5-0.5B on ECQA..

Figure 15: Different kernel shows similar pattern.

## L  USE OF LARGE LANGUAGE MODELS.

Large language models (LLMs) were used solely as assistive tools for proofreading and improving clarity of writing.

