# OpenReview forum: "The Generalization Ridge: Information Flow in Natural Language Generation"
_ICLR.cc/2026/Conference — Submitted to ICLR 2026_

### Official Review · Reviewer_uQHu · 2025-10-31

**Soundness:** 4
**Presentation:** 3
**Contribution:** 4
**Rating:** 8
**Confidence:** 3

**Summary:**

This work analyzes the representations across transformer layers in a scenario of fine-tuning for natural language generation tasks. The authors propose an approach based on mutual information between the internal model representations and the ground truth target token, and formalize metrics of "predictive information" and "incremental information gain", corresponding to a specific layer hidden state or a specific layer residual, respectively. These mutual information metrics rely on matrix-based Renyi entropy. In a series of analysis experiments the work demonstrates a behavior termed "generalization ridge", where the mutual information with the task label reaches a peak before the final transformer layer, and this point in the model also corresponds to better out-of-domain generalization performance.

**Strengths:**

1. Some very interesting findings on model behavioral dynamics, that are consistent across models and tasks.
2. The conclusions are supported through multiple complementary experimental setups: the observational results are complemented by intervention experiments - controlling the contribution of each transformer layer via residual scaling coefficients; standard NLG data is used alongside a controlled synthetic dataset setting; mutual information results are complemented by an explicit analysis of attention to signal-bearing tokens.

**Weaknesses:**

1. My feeling is that the interpretation of the data is probably a bit over-simplified, given that high mutual information between a representation and the target label can also be potentially consistent with memorization. Throughout the paper a rise in predictive information is interpreted as increased generalization, whereas in the overfitting experiment (l. 332) a similar rise is described as "memorizing shortcuts" and "redundant label-specific noise". This is interesting, and helps to give the reader a more complete picture, but also highlights that the meaning of the predictive information metric may not be as straightforward as the authors make it out to be.
2. The analysis is limited to fine-tuning scenarios. I think it would have been informative to also see some examples of these MI metrics in a zero-shot/few-shot scenario.

**Questions:**

1. For Table 1 accuracy results, how exactly is the early-exit implemented? I see that the language modeling head uses the same weights as for the input embedding, but still - maybe the results are influenced by a mismatch between the language modeling head and the hidden states of intermediate layers? AFAIU the embedding/unembedding matrix weights are only optimized for projecting the input and last layer output.
2. Is the approach for creating the synthetic dataset new? Or did other works use similar approaches?

Minor suggestions:
* I felt Figure 1 in its current form is not particularly easy to follow visually, and thus less informative for the reader. I suggest to think a bit about how to organize it visually and create a stronger connection between the text and the graphics.
* The transition to the part about "signal attention" (l. 343-345) can be improved, I found it a bit confusing. My suggestion is to replace that with the more explicit goal of this experiment (to quantify where in the network attention to semantically important information is concentrated, as currently stated in l. 350) and then describe the experiment.

Typos:

l. 125 reflects -> that reflects

l. 328/330 generalization ridge -> a/the generalization ridge

l. 401 blocks -> block

l. 921 quantity -> quantifies, blocks -> block

---

> ### Author Response · Authors · 2025-12-03
>
> We sincerely thank the reviewer for the positive assessment and are encouraged that the **core findings** and the **complementary experimental design** were recognized. We address the reviewer’s thoughtful questions and clarifications below.
>
> **Regarding the interpretation of mutual information**
>
> We thank the reviewer for finding the phenomenon interesting! Our interpretation focuses on **where** and **when** predictive information appears, combined with multiple behavioral signals: the layers with the predictive-information peak also achieve the strongest OOD performance (Table 1), the highest attention to task-relevant tokens (Table 2), and are upweighted under distribution shift in the residual-scaling probe (Figure 6). Any single measurement may be ambiguous, but taken together, the evidence is highly convergent.
>
> Moreover, the distinction between generalization and memorization becomes clearer when considering **training dynamics**. Before overfitting, the rise of predictive information in mid layers reflects that the model is first acquiring the useful, abstract, generalizable components of the label. During this phase, the top layers largely behave as ***pretrained decoders***—focusing on surface-level patterns such as local token co-occurrence, syntactic templates, or corpus-specific biases. These signals are not strongly aligned with the task label, so late-layer I(Z_\ell;Y) remains relatively low.
>
> After overfitting, however, the later-layer rise in predictive information reflects a **different mechanism**: the model begins to encode redundant or instance-specific label patterns, effectively shifting from generalization to **memorization**. Thus, the predictive-information surges occur at **different depths** and **different phases** of training, revealing distinct functional roles: mid layers tend to encode abstract, decision-relevant structure, whereas late layers increasingly specialize toward task-specific or even sample-specific patterns.
>
> Seen this way, the overfitting experiment (Figure 4) actually **strengthens** our interpretation rather than contradicting it. The key insight is not the absolute MI value but **where, when, and how** predictive information manifests. This layered division of labor—and the contrasting behaviors of mid- vs. late-layer information—is precisely the phenomenon our paper aims to highlight.
>
>
> **Regarding the fine-tuning scenario**
>
> Our goal is to understand how the generalization ridge forms and evolves **during training**, so our analysis centers on fine-tuning and observing the dynamics of information accumulation and reorganization. To complement this, we also provide a **zero-shot GSM8K evaluation** in Appendix J, showing that the **similar pattern** as intermediate-layer predictive-information peaks.
>
> For early exiting, we apply the shared input/output embedding matrix to all layers and our current setup already shows a clear pattern: intermediate layers achieve higher OOD accuracy than the final layer. This supports our central claim without requiring any additional heads. If we trained a layer-specific projection head for each layer, intermediate-layer accuracy would only increase because a dedicated head can always better fit that layer, which would further strengthen rather than weaken the observed trend. Besides, weight tying between the input and output embeddings further reduces latent-space mismatch and is the standard practice for early-exit evaluation in autoregressive LMs. Overall, our setup provides a conservative estimate of each layer’s predictive ability, and supports our claim.
>
> For synthetic dataset, the dataset is new, though inspired by classic synthetic tasks (modular arithmetic, structured noise). We have revised the paper to improve clarity and correct the typos.

---

> ### Author Response · Authors · 2025-12-03
> **Response Summary**
>
> We clarify how predictive information should be interpreted across training and explain our experimental setup.
>
> (1) Interpreting mutual information. Our interpretation focuses not on absolute MI values but on **where** and **when** predictive information emerges. The intermediate-layer peak consistently aligns with multiple behavioral signals—strongest OOD accuracy, highest attention to task-relevant tokens, and selective upweighting under distribution shift.
>
> Training dynamics clarify the meaning of these peaks: before overfitting, mid-layer rises reflect the formation of abstract, generalizable structure, while top layers still reflect pretrained surface patterns. After overfitting, late-layer rises instead indicate memorization of label-specific noise. These rises occur at different depths and stages, revealing a clear depth-wise division of labor at the heart of the generalization ridge.
>
> (2) Experiment setup. We analyze fine-tuning because our goal is to track how the generalization ridge forms and evolves **during learning**. A complementary zero-shot GSM8K evaluation shows the same intermediate-layer peak.

---

### Official Review · Reviewer_MQbN · 2025-10-31

**Soundness:** 2
**Presentation:** 3
**Contribution:** 2
**Rating:** 2
**Confidence:** 4

**Summary:**

This paper analyzes the mutual information between hidden layers and outputs for language models across a few tasks. The paper finds a spike in the upper middle layers which they coin generalization ridge.

**Strengths:**

- The paper is a nice application of using information theory to identify layer importance
- The finding is self-contained and the paper fairly easily to follow

**Weaknesses:**

At a high level, I was disappointed after reading the paper for few reasons. (1) I was missing the reason for why I should care about these results, especially since the results are not connected to the existing body of literature. There are many prior papers that investigate layer-wise importance and the result that upper middle layers are important is not new. (2) The key claims even in the title relate to shedding light into NLG. But, there is no actual experiment on NLG tasks and the experiments only look at a single output, not even multiple tokens.

**Some related work**: Some discussions I was expecting to see were grounding in work like "BERT rediscovers the classical NLP pipeline" which relates layers to specific NLP tasks (plus an extensive body of literature around this). I was expecting a discussion of causal vs. correlational probing work, for example causal mediation analysis as an alternative way to identify layer/neuron importance. And since this was investigated, I was expecting more relation to work on training dynamics.

Similarly, there is a lot of prior work on information-theoretic probing approaches that the method here does not warrant a new name. This is an implicit overclaim.

**Lack of Relation to NLG**

The model selection is very surprising and limited to very small models. The older models here are not models that were ever used for actual NLG work.
Moreover, none of the datasets are well known NLG datasets or classic NLG tasks (such as summarization, dialog response generation, or surface realization). The arithmetic dataset is not at all an NLG dataset. In addition, looking at a single output representation implicitly looks at only a single output and not a longer utterance as would be expected in NLG.

**Relationship between token representation and output**
The extraction of $\Delta$Z (the change in token representation) implies that the information is represented in linear space. While this could be a helpful approximation of the ground truth, this is a very strong assumption to make, given the very much non-linear nature of the model. Moreover, page 4 states that delta Z is further $l2$ normalized which further changes the nature of the representation away from what the model represents

On a clarity note, I don’t understand why the Matrix-Based Mutual Information section does not continue to use the prior representation introduced above. This seems rather lazy as it is left to the reader to connect how it is applied. Not even the information in the appendix discusses details

**Questions:**

n/a

---

> ### Author Response · Authors · 2025-12-03
> **Response (1/2)**
>
> We thank the reviewer for their assessment and appreciate the positive comments on clarity and on the value of information-theoretic analysis. Below, we address the reviewer’s concerns regarding related work and NLG setting, and we clarify the technical misunderstandings raised.
>
> **Regarding related works**
>
> Regarding related work, we emphasize that Tenney et al. characterize which **linguistic tasks** each layer correlates with in **encoder-only** models. It identifies clusters of linguistic skills but does not analyze how downstream task-relevant information evolves, nor do they connect the results with the generalization vs. memorization behavior across the depth of an autoregressive transformer. In contrast, our work directly tracks how information is accumulated, abstracted, and later overwritten by memorized patterns in top layers, revealing the non-monotonic “generalization ridge”—a phenomenon that task-specific probes cannot detect because they regress onto external labels rather than measuring the model’s own predictive signal. Prior probing methods are mostly correlational analysis: they read out information without changing the model’s computation. By comparison, our residual-scaling mechanism performs an actual intervention on the residual pathway, allowing the model itself—**not an external classifier**—to adjust reliance on each block. This functional probing makes it possible to causally test how models tend to allocate the functions of each layer. Finally, unlike prior work that only inspects learned representations at convergence, our paper explicitly analyzes **training** dynamics, showing how information evolves throughout training. Our paper also cites training-dynamics work such as Merchant et al. which analyze fine-tuning in BERT. We also include more related work for training dynamic and discussions for casual analysis in the revised paper.
>
> **Regarding the NLG and Multiple Token Generation**
>
> Our work examines how autoregressive language models organize and propagate information during next-token prediction. While the benchmarks we study are not traditional long-form NLG tasks such as summarization or translation, they are executed entirely through an **autoregressive next-token generation mechanism**. In all cases, the model receives a natural-language prompt and predicts the next token from the full vocabulary distribution. Our analysis focuses precisely on this generative computation, which is shared across all autoregressive LLM usage scenarios. Because our analysis concerns how hidden representations evolve during this generative decoding, the information-flow patterns we measure directly reflect the model’s natural-language generation process. We include a synthetic dataset for a more controlled setting to better differentiate ID and OOD scenarios.
>
> Importantly, multi-token generation is fundamentally a sequence of single-token predictions; the information-flow process does not change, but is applied repeatedly across positions. We would like to emphasize that understanding the single-token case is **foundational** to understand multi-token behavior and the single-step analysis can be **easily extended** to multi-token scenarios.
>
> We provide a multi-token analysis on a classic summarization task (CNN/Daily Mail)  in Appendix I, which shows that the **same ridge structure persists** across extended generations. We examine multi-token generation by jointly characterizing how layer representations shape the model’s predictive direction and how they align with the gold-answer (details in Appendix I). The results reveal a coherent pattern: a stable and pronounced information ridge centered in the intermediate layers across steps. This indicates that intermediate layers concentrated most meaningful and decision-relevant information. In contrast, upper layers exhibit behavior that might be related to pretrained surface patterns from corpus such as token co-occurrence, causing the representation to drift away from this decision-relevant structure. A notable trend is the first generation step, which shows the sharpest ridge because it compresses the full prompt into a single initial decision; after this global transition, subsequent steps follow a more regular autoregressive pattern, producing flatter but still ridge-centered profiles. Overall, the analysis consistently points to intermediate layers as the locus of generalizable, decision-relevant structure, whereas upper layers increasingly reflect surface-level patterns learned during pretraining.

---

> ### Author Response · Authors · 2025-12-03
> **Response (2/2)**
>
> **Clarifying the technical misunderstanding**
>
> The reviewer’s comment is based on a misunderstanding of our method.
>
> ΔZₗ is not a linearization. It is the exact residual update produced by the Transformer block:
>
> [**zₗ = zₗ₋₁ + blockₗ(zₗ₋₁) ⇒ Δzₗ = zₗ − zₗ₋₁ = blockₗ(zₗ₋₁)**]
>
> We **do not** approximate or linearize the model in any way; we simply read the model’s intrinsic residual update.
>
> Our mutual-information estimator also does not assume linear structure. Matrix-based entropy uses kernel Gram matrices and is non-linear. The normalization applied before constructing Gram matrices is a standard preprocessing step for kernel-based information measures. It removes only the global magnitude and stabilizes entropy estimation. It does not alter the representational geometry or impose linearity; it simply ensures that the kernel method behaves properly.
> In short, computing ΔZ_l is reading out the exact residual update performed by the model, **not** a linearization assumption. The reviewer’s concern does not apply to our method.
>
> **Regarding the paper writing**
>
> Section 3 clearly states we estimate I(Z(ℓ);Y) and I(ΔZ(ℓ);Y) using Equations 1–2, where U and V are instantiated as Z(ℓ) or ΔZ(ℓ) and Y. We intentionally present the estimator in its general form because the computation is reused for both quantities, and repeating the notation would be redundant.
>
> On the Method Name, InfoRidge is not just "information-theoretic probing", it is a framework combining: Matrix-based MI estimation adapted to autoregressive LMs in NLG, dual metrics (predictive information + incremental gain) and training dynamics analysis. The name reflects this integrated approach, not only for MI estimation itself.

---

> ### Author Response · Authors · 2025-12-03
> **Response Summary**
>
> We clarify the NLG setting, correct the reviewer’s technical misunderstanding, and distinguish our work from prior probing literature.
>
> (1) NLG framing. While our datasets are not classic long-form NLG benchmarks, all experiments are executed through **autoregressive next-token generation**. The model receives a natural-language prompt and predicts each token from the full vocabulary. Our analysis targets this generative decoding process directly. We would like to emphasize that understanding the single-token case is **foundational** to understand multi-token behavior and the single-step analysis can be **easily extended** to multi-token scenarios. We added Appendix I to show that the same ridge structure appears in multi-token summarization using CNN/Daily Mail which is a classic summarization benchmark.
>
> (2) Clarifying the misunderstanding. The reviewer’s claim of “linearization” is incorrect:
>
> zₗ = zₗ₋₁ + blockₗ(zₗ₋₁) ⇒ Δzₗ = zₗ − zₗ₋₁ = blockₗ(zₗ₋₁),
>
> which is the exact residual update performed by the model. Our mutual-information estimator is also nonlinear, and the normalization only removes global magnitude for stabilizing entropy estimation and does not alter the representational geometry. Thus, our method relies on the model’s intrinsic computations and **does not** rely on any linearizing assumptions.
>
> (3) Relation to prior work. Prior probes (e.g., Tenney et al.) analyze encoder-only models and measure **how well layers encode external semantic labels**. They provide correlational snapshots but do not track how an autoregressive LM’s own predictive information evolves or how layers shift from generalization to memorization. In contrast, InfoRidge measures the LM’s next-token predictive signal and its incremental gain, yielding a training-aware and causal view that exposes the generalization ridge, which semantic feature probes cannot see.

---

### Official Review · Reviewer_fBSE · 2025-10-31

**Soundness:** 2
**Presentation:** 3
**Contribution:** 2
**Rating:** 4
**Confidence:** 3

**Summary:**

This paper investigates how Transformers operate internally and how task-specific generalization emerges during training and propagates across layers for natural language generation. The authors estimate mutual information between layer-wise representations and the outputs, and learn per-residual-block scaling coefficients to assess each layer’s contribution. Using these coefficients, they find that mid-layers become substantially more important under distribution shift, whereas later layers are more tied to memorization of in-distribution patterns.

**Strengths:**

1) The paper studies very important questions of how task-specific generalization abilities emerge in a transformer layer, which is largely not understood.
2) The paper is well written, easy to follow and the core points are well articulated.

**Weaknesses:**

1) It’s unclear how “which layers are important” alone explains how generalization emerges. The study doesn’t identify the causal factors behind these abilities.

2) Recent work already reports that middle (and later) layers matter, and that final layers can be poor for OOD generalization (e.g., [1], which is cited but not discussed much). Can the authors discuss more how the new findings from these papers are different the existing ones to understand how generalization emerges and what are the practical usefulness of these findings.

3) The evaluation datasets are limited. It would be valuable to relate the identified “ridge” layers to varied task types—math, general reasoning, and multi-hop reasoning. Please add results on standard datasets such as MMLU, GSM8K, HotpotQA etc.

4) There is no accompanying theoretical analysis.

[1] Uselis, Arnas, and Seong Joon Oh. "Intermediate layer classifiers for ood generalization." arXiv preprint arXiv:2504.05461 (2025).

**Questions:**

1) Is there a specific reason the authors use a Gaussian kernel for entropy estimation? Could they include an ablation with alternative kernels? I want to see the sensitivity of the results depending on the kernel choice.

2)  Although multiple model sizes are considered, the scaling trends are not clearly characterized. Can you clarify how layer importance evolves with width/depth. I want to see the evaluation on existing open source models, no training is needed.

3) It seems the models are fine-tuned from pretrained checkpoints, and the layer-wise mutual information “peakedness” seems to reflect pretrained biases. Do the claims about ridge layers, memorization, and generalization still hold when training from scratch?

4) Suggestion: Consider splitting Figure 2 into multiple panels/figures to improve readability.

---

> ### Author Response · Authors · 2025-12-03
> **Response (1/2)**
>
> We thank the reviewer for the thoughtful feedback and for recognizing the **importance** and **clarity** of our work. We address all concerns with clarifications and new analyses.
>
> **Regarding the causal factors behind generalization**
>
> We first empirically observe a clear relationship between predictive information and the model’s generalization–memorization behavior. Beyond information dynamics, our additional analyses provide indications of **why** generalization emerges. The attention study (Fig. 3) suggests that generalizable structure may arise because mid-layers **concentrate attention on signal tokens**, while later layers shift toward last-token attention patterns associated with memorization. In parallel, the model-capacity analysis (Table 2) shows that the ridge only appears once the network exceeds a critical depth, suggesting that redundant or sufficient **capacity** may be another enabling factor for the formation of generalizable representations. Together, these observations suggest that evolving attention patterns and capacity-driven representational structure provide plausible mechanisms for how generalization emerges.
>
> **Regarding the difference between our paper and the paper [1] mentioned by reviewer**
>
> We emphasize that the problem setting, scientific questions and methodology are fundamentally different.
>
> i) Our paper focuses on the NLG task rather than the vision task.
>
> Uselis & Oh study vision models under OOD image classification shifts. Their ILCs are linear classifiers trained on frozen intermediate features. In contrast, our work analyzes **autoregressive language models** trained for **next-token generation**, where every layer processes different information. The generalization behavior in autoregressive LMs is structurally different from vision models.
>
> ii) e track **predictive information propagation** through **training** and uncover the **mechanisms** behind generalization rather than merely identifying a good OOD classifier.
>
> Uselis & Oh ask which intermediate representation yields the best OOD classifier. Their method does not analyze **training dynamics**, or information propagation. Our work tracks how predictive information evolves throughout training and further explains why mid-layers support generalization by analyzing the autoregressive **attention patterns** that develop as the model learns.
>
> iii) Our work provides **causal** analysis.
>
> ILCs are purely observational probes. We introduce residual scaling coefficients as trainable indicators of how the model itself reallocates representational focus across layers during training. This allows us to move beyond static or correlation-based analyses by capturing how the model internally adapts and adjusts as it learns.
>
> **Regarding the experimental details**
>
> We want to clarify that the datasets we use already cover fundamentally different tasks: Synthetic Arithmetic is an arithmetic task, CLUTRR is a multi-hop relational reasoning benchmark and ECQA commonsense QA benchmark. Furthermore, to address the reviewer’s concern, we additionally include a **non-training GSM8K evaluation** on Qwen (Appendix J). The model still exhibits a clear ridge pattern, demonstrating that the phenomenon persists across both trained and non-training settings. Our primary analyses focus on trained models because our motivation is to understand how generalization emerges during training.
>
> The matrix-based MI estimator requires a positive-definite Gram matrix. We use the Gaussian kernel because it is the standard, widely adopted choice that satisfies this requirement. We also tested the effect of alternative kernels including polynomial kernel and laplacian kernel, and the results, which showed robustness to kernel choice, are included in Appendix K.
>
> **Regarding the theoretical background**
>
> We would like to note that we provide theoretical grounding in Section 3 and Appendix A.3: We connect to the Min Wasserstein Generalization Bound (He et al. 2024), which proves that generalization error is minimized at the layer where representations optimally align with the true data distribution—the "generalization funnel."

---

> ### Author Response · Authors · 2025-12-03
> **Response (2/2)**
>
> **Regarding the trend summarization**
>
> Across all models, we observe a consistent scaling trend: depth predominantly determines whether and where the generalization ridge emerges. Within the same model family, such as the GPT-2 Small and Medium variants, differing in both width and depth, exhibit a similar relative ridge position, showing that the information dynamics are stable across scales within a family. This suggests that width is likely not the primary factor governing where the ridge occurs.
>
> **Regarding the pretrained influence**
>
> We focus on pretrained LLMs because this is the regime in which modern NLG systems operate, and we want to analyze models that have already internalized the fundamental structure of natural language. Pretraining inevitably shapes layer-wise representations, for example through corpus-specific pattern memorization. In our view, the emergence of the ridge reflects both architectural factors (e.g., how residual blocks distribute computation) and these pretrained biases from corpus, and it is sharpening during fine-tuning on task-specific learning.

---

> ### Author Response · Authors · 2025-12-03
> **Response Summary**
>
> We clarify the causal interpretation of our findings and strengthen the empirical evidence.
>
> (1) Why generalization emerges.
> We observe a consistent link between predictive information and generalization–memorization behavior, and our **attention** and **capacity** analyses explain why the ridge forms: mid-layers selectively attend to task-relevant signal tokens, and the ridge only appears once the model has sufficient representational capacity to form.
>
> (2) Distinct contributions.
> Unlike the work the reviewer mentioned, our study analyzes autoregressive LMs trained for **next-token generation** rather than vision models, and we track **predictive information propagation** through training and uncover the mechanisms behind generalization rather than merely identifying a good OOD classifier. Moreover, our causal residual-scaling experiment shows that the model itself reallocates layer usage under ID–OOD shift—providing causal evidence of functional layer organization beyond static correlational probes.
>
> (3) Empirical and theoretical stability. We add a no-training GSM8K evaluation on Qwen, which exhibits the same ridge phenomenon. We also clarify the theoretical grounding. Together, these results show that the ridge pattern is both empirically robust and theoretically supported.

---

### Official Review · Reviewer_4ebs · 2025-11-02

**Soundness:** 3
**Presentation:** 4
**Contribution:** 3
**Rating:** 6
**Confidence:** 3

**Summary:**

This paper introduces InfoRidge, an information-theoretic framework to analyze how predictive information flows through transformer layers during natural language generation (NLG). Using matrix-based mutual information estimation, the authors track two key quantities: 1) mutual info between hidden states at layer $l$ and target token (predictive information) and 2) info added by the residual update at layer $l$ (incremental information gain). Across GPT-2, Qwen-2.5, and LLaMA models on CLUTRR, ECQA, and a synthetic arithmetic dataset, the authors consistently observe a non-monotonic “generalization ridge”: predictive info peaks in upper-middle layers then declines, while final layers appear to memorize. They corroborate this with learnable residual scaling coefficients, allowing the model to assign layer importances depending on the layer influence towards prediction. Authors find that under distribution shifts, models automatically down-weight final layers and up-weight ridge layers, providing further evidence to the existence of generalized intermediate representations. The ridge emerges only beyond a depth threshold, and attention/visualization analyses show ridge layers focus on task-relevant tokens whereas final layers revert to positional heuristics.

**Strengths:**

1.	Originality: The proposed adaptation of matrix-based mutual information to next-token prediction is an interesting idea to identify layer influence towards predictions.
2.	Quality: Authors conduct extensive experiments across model families & tasks, careful controls (depth ablation, overfitting, attention/decoder visualizations).
3.	Clarity: Hypotheses, metrics, and takeaways are explicitly stated; additional mathematical details are added to appendices.
4.	Significance: The proposed method based on mutual information can provide a principled lens to locate “where” generalization happens, immediately useful for early-exit, model pruning, or targeted fine-tuning strategies.

**Weaknesses:**

1. **Scalability of MI estimation**: In Appendix C, the authors state that 50-200 examples are subsampled to avoid excessive memory usage from large Gram matrices. Provided authors experimented mainly with small language models, this raises doubts on the applicability of such a method to analyze larger state-of-the-art LLMs.
2.	**Lack of proper causal verification**: While the analysis shows that ridge layers correlate with better OOD accuracy, the use of residual scaling coefficient to test their importance in a task-directed fashion might lead to confusing results, since 1) $beta$ coefficients are trained after full fine-tuning with frozen weights; it is unknown whether jointly learning $beta$ from scratch preserves the ridge or changes training dynamics; and 2) subsequent layers depend from previous ones in their computation, creating an implicit incentive to upweight earlier layers to reduce information loss. A more convincing analysis could, for example, employ activation patching methods to localize the units that contribute most to model predictions in OOD settings, showing that these are mainly contained within ridge layers.
3.	**Findings are not particularly novel**: While the characterization of the middle-to-final layers as responsible for generalization is novel, there is ample literature showing that these layers perform best when used for probes aiming to extract semantic task-related information (see e.g. the early work "BERT Rediscover the Classical NLP Pipeline" by Tenney et al., showing a hierarchy of linguistic information within BERT peaking at around 2/3rds of the model). While the analyses conducted by the authors are undoubtedly of value in providing additional evidence for the semantic generality of such layers, this is unlikely to affect the current practices that already employ those layers for interpretability analysis on language models.

**Questions:**

In the early exiting results shown in table 1, are different prediction heads trained over a frozen model, or is the final head simply applied to the selected layer? In the latter case, the mismatch between latent spaces in absence of a projection could lead to misleading results.

---

> ### Author Response · Authors · 2025-12-03
> **Response (1/2)**
>
> We thank the reviewer for the thoughtful and constructive feedback. We are encouraged that the reviewer recognizes the **originality** of adapting matrix-based mutual information to next-token prediction, the **quality** of our experiments and controls, the **clarity** of our paper, and the **significance** of our framework. Below we address the reviewer’s concerns on scalability, causal verification, and novelty on findings.
>
> **Regarding the Scalability of MI Estimation**
>
> Our method is highly **memory-efficient** and fast and this is actually an advantage of matrix-based MI estimation over traditional approaches.
>
> The memory complexity is O(N² + d) for N samples with hidden dimension d. For example, with LLaMA-3.1-8B (d=4096) and N=100 samples, our method requires only ~2MB per layer. This makes it highly scalable to modern large language models.
> The small sample range is intentionally chosen for efficiency, not imposed by limitations. Small samples are sufficient to capture stable population-level information patterns.The results are stable across five random seeds, and we report 96% confidence intervals, demonstrating that small samples are sufficient and can get stable population-level information patterns. The method is highly practical for analyzing modern LLMs.
>
> **Regarding the causal verification**
>
> Our analysis focuses on how the model reorganizes its computation after learning the task. Before fine-tuning, the pretrained model does not encode any ID–OOD distinction for a specific task, so the scaling cannot be meaningfully interpreted at that stage. For the second concern, this is actually our motivation for this experiment design. The residual scaling experiment traces how the model itself dynamically learns and allocates which layers to use during training. This considers meaningful inter-layer interactions and reveals how the model adaptively reweights layer contributions in response to generalization demands or distribution shifts.
>
> Following the reviewer’s suggestion, we additionally perform a controlled activation-patching experiment to directly test the role of each layer in OOD prediction (Appendix H). For each OOD sequence, we pair it with an ID sequence of identical structure and identify the last context token. At layer l, we run a patched forward pass where we replace this token’s hidden state in the OOD run with the hidden state at the same token position from the ID run, while keeping all other activations unchanged. We then measure the drop in the correct-token logit, which isolates the contribution of layer l. The results reveal an aligned result with our information ridge. Layers 1–9 yield negative Δₗ values, meaning that OOD activations in these layers are unhelpful or even harmful—replacing them with ID activations increases the correct-token logit. In contrast, layers 10–12 exhibit large positive Δₗ values, and and critically, the effect rises sharply at layer 10, peaks at layer 11, and decreases again at layer 12. This **rise–peak–fall pattern mirrors the ridge behavior** observed in Figure 2, providing direct causal support that the ridge layers are in fact necessary for the model’s generalization behavior. Below we show the logit drop Δₗ (baseline logit − patched logit). Larger values indicate greater contribution to OOD prediction.
>
> | Layer ℓ | 1        | 2        | 3        | 4        | 5         | 6         | 7         | 8          | 9        | **10**   | **11**    | **12**    |
> |--------|----------|----------|----------|----------|-----------|-----------|-----------|------------|----------|----------|-----------|-----------|
> | Δₗ     | -0.1037  | -0.1039  | -0.0327  | -0.5405  | -5.1663   | -8.3992   | -5.8914   | -15.0996   | -3.0998  | **9.3148** | **17.2022** | **15.5943** |

---

> ### Author Response · Authors · 2025-12-03
> **Response (2/2)**
>
> **Regarding novelty of findings**
>
> We sincerely thank the reviewer for the thoughtful feedback and the reference to Tenney et al.'s work. However, we respectfully believe our work addresses fundamentally different questions with distinct contributions.
>
> Our work addresses a **fundamentally different** question from **BERT-style linguistic probing**. Rather than locating static syntactic or semantic features, we analyze training-time predictive information dynamics in modern autoregressive LMs and connect these dynamics to generalization versus memorization behavior.
>
> 1. *We study generation on modern Autoregressive LMs instead of BERT on classification with discrete linguistic features.*
> **Autoregressive generation** involves fundamentally different information flow due to causal attention and continuous next-token prediction, which requires different analytical approaches.
>
> 2. *We focus on how predictive information evolves during training and their connection to generation vs. memorization behavior.*
> Tenney et al. ask "What linguistic features (syntax, semantics) are encoded in which layers?”
> Our work asks "How does information evolve during training, and how do different layers functionally support generalization vs. memorization?"
> Unlike prior probing studies that infer semantic features through external classifiers, our framework directly measures the model's **own predictive signal**—I(Zℓ; Y) and I(ΔZℓ; Y)—across layers and throughout training.
>
>    This reveals: *(i) a consistent three-phase trajectory where predictive information peaks at intermediate layers, forming a generalization ridge; (ii) that ridge layers contribute the largest incremental information gain, identifying where task-relevant structure is introduced; (iii) training dynamics show distinct information trajectories before and after overfitting; and (iv) under distribution shift, models will automatically reallocate layer contributions—downweighting final layers and upweighting ridge layers.*
>
> 3. *Our analysis goes beyond observation to explain **why** generalizable structure concentrates at intermediate layers.*
> At the attention level, we find that attention to task-important tokens peaks precisely at the ridge layers, while later layers increasingly shift toward last-token attention—a pattern indicative of shortcut behavior and memorization. At the capacity level, we show that the ridge emerges only once the model exceeds a critical depth, revealing that sufficient representational capacity is required for intermediate layers to develop robust, generalizable structure.
>
> 4. Operational Contributions.
> We provide:
> i) InfoRidge framework: Information-theoretic quantification tailored for autoregressive generation
> ii) Causal validation: Learnable residual coefficients demonstrate models actively adapt layers under distribution shift. This provides causal evidence for functional layer organization, which is beyond correlational probing.
>
> For early exiting, we apply the shared input/output embedding matrix to all layers and our current setup shows a clear pattern: intermediate layers achieve higher OOD accuracy than the final layer. This supports our central claim without requiring any additional heads. If we trained a layer-specific projection head for each layer, intermediate-layer accuracy would only increase because a dedicated head can always better fit that layer, which would further strengthen rather than weaken the observed trend. Furthermore, weight tying between the input and output embeddings further reduces latent-space mismatch and is the standard practice for early-exit evaluation in autoregressive LMs.

---

> ### Author Response · Authors · 2025-12-03
> **Response Summary**
>
> We address reviewer concerns with new analyses and clarifications.
>
> (1) Scalability. Our matrix-based MI estimator is extremely **lightweight** (O(N²+d)) and scales to modern LLMs. The small-sample regime is intentional for efficiency instead of a limitation, and five-seed 96% CIs show stable population-level patterns.
>
> (2) Causal verification. Our focus is on how the model reorganizes computation after learning the task. The residual-scaling analysis shows that the model itself **dynamically reallocates** layer usage under ID–OOD conditions, revealing functional distinctions across depth. Following the reviewer’s suggestion, we additionally performed a controlled activation-patching experiment. The results exhibit the **same rise–peak–fall structure** as our MI ridge, providing supporting causal evidence that the ridge layers are necessary for generalization.
>
> (3) Novelty of findings. Our work addresses a **fundamentally different** question from **BERT-style linguistic probing**. Rather than locating static syntactic or semantic features, we analyze training-time predictive information dynamics in modern autoregressive LMs and connect these dynamics to generalization versus memorization behavior. This perspective uncovers several findings: a consistent three-phase information trajectory with a pronounced intermediate-layer ridge; the identification of ridge layers as the dominant source of incremental information gain; distinct information regimes before and after overfitting; and an adaptive redistribution of layer contributions under distribution shift. These results reveal not only what structure emerges, but also why intermediate layers concentrate generalizable information.

---

### Author Response · Authors · 2025-12-03
**Summary for the AC (2/2)**

**(3) Clarifying the technical misunderstanding.** (See details in responses to reviewer MQbN.)

We want to clarify that reviewer MQbN’s concern about “linearization” is based on a **technical misunderstanding**. By definition,
zₗ = zₗ₋₁ + blockₗ(zₗ₋₁) ⇒ Δzₗ = zₗ − zₗ₋₁ = blockₗ(zₗ₋₁),
so ΔZₗ is the exact residual update produced by the model, not a linear approximation. Our mutual-information estimator is also nonlinear, and the normalization only removes global magnitude for stabilizing entropy estimation and does not alter the representational geometry. *Thus, our method relies on the model’s intrinsic computations and **does not** rely on any linearizing assumptions.*

We also address the following minor concerns:

1. Added activation-patching experiment (Appendix H); results show a rise–peak–fall pattern mirroring the information ridge, providing support for the role of ridge layers in OOD generalization. (4ebs)

2. Added kernel ablations with Laplacian and polynomial kernels (Appendix K); the ridge location and trend remain consistent. (fBSE)

3. Added non-finetuned evaluation on GSM8K (Appendix J), showing similar mid-layer ridge. (fBSE, uQHu)

4. Clarified theoretical grounding; Section 3 and Appendix A.3 connect our framework to the Min Wasserstein Generalization Bound, which provides a theoretical foundation to our work. (fBSE)

5. Clarified experimental details: our MI estimator is memory-efficient and statistically stable; early-exit uses a conservative setup that does not weaken our claims. (4ebs, uQHu)

---

### Author Response · Authors · 2025-12-03
**Summary for the AC (1/2)**

We sincerely thank the reviewers and the AC for taking the time to review our paper and for their valuable comments and feedback. We are encouraged that reviewers recognized: (1) the **novelty and importance** of using matrix-based information-theoretic approach to study how generalization emerges across transformer layers during autoregressive NLG (4ebs, fBSE); (2) the **clarity** of our presentation (4ebs, fBSE, MQbN, uQHu); and (3) the **quality** of experiments, with consistent findings across models and tasks supported by multiple complementary analyses (uQHu, 4ebs).

We have addressed and clarified the four main concerns raised by reviewers:

**(1) Contribution and novelty.** (See details in responses to reviewers 4ebs, fBSE, MQbN.)

Our work introduces a quantitative framework that reveals how generalizable information emerges inside autoregressive natural language generation models and distinguishes it from later-layer memorization.

(i) **Novelty of Methodology: An information-theoretic framework for autoregressive NLG**

We introduce InfoRidge, an information-theoretic framework that characterizes layer-wise information flow in autoregressive generation. By jointly estimating predictive information and incremental information gain, InfoRidge reveals how transformers acquire, refine, and compress task-relevant information throughout training. It provides a *principled analysis that quantifies the model’s predictive information evolution during training in the NLG mechanism*.

(ii) **Novelty of Operationalization: Residual scaling coefficients as causal layer-function probes**

We propose a **novel causal probe**, residual scaling coefficients, that learns layer-wise residual gates while keeping all model weights frozen. This design isolates each block’s functional contribution under distribution shift and demonstrates, causally, how models downweight memorization-heavy late layers and shift reliance toward intermediate layers that support generalization. This allows us to move beyond static or correlation-based analyses by **capturing how the model internally adapts and adjusts as it learns to fit OOD distribution**.

(iii) **Novelty of Findings: Information Dynamics and the Generalization Ridge**

**Unlike prior probing studies that infer semantic features through external classifiers**, our framework addresses a **fundamentally different** question: it directly measures the model's own predictive signal—I(Zℓ; Y) and I(ΔZℓ; Y)—across layers and throughout training. This reveals: *(i) a consistent three-phase trajectory where predictive information peaks at intermediate layers, forming a generalization ridge; (ii) that ridge layers contribute the largest incremental information gain, identifying where task-relevant structure is introduced; (iii) training dynamics show distinct information trajectories before and after overfitting; and (iv) under distribution shift, models will automatically reallocate layer contributions—downweighting final layers and upweighting ridge layers.*

(iv) **Contribution to understanding underlying causes**
Our analysis goes beyond observation to explain why generalizable structure concentrates at intermediate layers.

- Attention-level evidence, showing that *attention to task-important tokens peaks precisely at the ridge layers*, while later layers shift toward last-token attention, a pattern associated with shortcut behavior and memorization,

- Capacity-level evidence, ​​demonstrating that the ridge only appears once the model exceeds a critical depth, suggesting that *sufficient representational capacity is required* for intermediate layers to develop generalizable structure.

**(2) Extension to multi-token natural language generation.** (See details in responses to reviewer MQbN.)

We want to emphasize that our experiments are conducted in the **autoregressive next-token generation setting**, which is the fundamental computational step underlying modern language models. In this setting, the model receives a natural-language prompt and predicts each token from the full vocabulary. **The single-step analysis can be easily extended to multi-token scenarios.**

To further address the reviewer’s concern, we additionally evaluate **multi-token generation on CNN/DailyMail (a classic summarization benchmark)**. The same intermediate-layer ridge appears across extended generations, *confirming that the phenomenon persists beyond single-token prediction and reflects a general property of NLG behavior.*

---

### Meta-Review · Area_Chair_GjsR · 2026-01-05

**Summary:**

Across reviewers, there was agreement that the paper is clearly written and studies an important question about layer-wise behavior in transformers. However, there are several concerns.

1. The primary issue is limited novelty. Multiple reviewers noted that the central finding, "intermediate layers being more important for generalization and OOD performance", is already well documented in prior probing, early-exit, and OOD generalization literature, and the proposed method does not lead to substantially new insights or implications.

2. The causal evidence is weak or indirect. Several reviewers point out "The study doesn’t identify the causal factors behind the findings."

3. Several reviewers highlighted a mismatch between the paper’s NLG framing and its experimental scope, as most evaluations focus on single-token prediction on reasoning or synthetic tasks rather than standard, real-world NLG benchmarks.

4. Experiments rely exclusively on relatively small, pretrained models and do not include training-from-scratch analyses.

5. Scalability concerns.

**Reviewer Concerns:**

During the rebuttal, the authors have addressed the several technical concerns and made some clalrification, however, the core issues remains unsolved.

The rebuttal provided clarifications on some concerns such as the mutual information estimator, kernel choice, and normalization. The authors also added additional experiments such as MULTI-TOKEN OUTPUTS,  activation-patching to further support the findings.

However, several key concerns that remain outstanding:

1. The rebuttal does not convincingly distinguish the main findings from prior work, despite the authors’ attempts to differentiate the proposed method from previous studies.

2. While additional analyses were added, they remain post-hoc and do not clearly identify the causal mechanisms.

3. The study still relies on relatively small models.

**Reviewer Scores:**

Reviewer 4ebs: Likely would have remained the same rating as the novelty and causal interpretation are not fully resolved.
Reviewer fBSE: Likely would have remained unchanged. The rebuttal does not substantially address the reviewer’s core concerns regarding limited novelty, and no theoretical analysis.
Reviewer MQbN: Linkely would have raised from 2 to 4 as the authors addressed the several concerns.
Reviewer uQHu: Unchanged.

---

### Decision · Program_Chairs · 2026-01-26

Reject